# Minimax Rates for Learning Pairwise Interactions in Attention-Style Models

**Shai Zucker**\*
Department of Applied Mathematics,
School of Mathematical Science
Tel Aviv University
`shaizucker@mail.tau.ac.il`

**Xiong Wang**\*
School of Mathematics
Sun Yat-sen University
Guangzhou, China
`xiongwang@ualberta.ca`

**Fei Lu**
Department of Mathematics
Johns Hopkins University
Baltimore, USA
`feilu@math.jhu.edu`

**Inbar Seroussi**
Department of Applied Mathematics,
School of Mathematical Science
School of Computer Science
Tel Aviv University
`inbarser@tauex.tau.ac.il`

## Abstract

We study the convergence rate of learning pairwise interactions in single-layer attention-style models, where tokens interact through a weight matrix and a nonlinear activation function. We prove that the minimax rate is $M^{-\frac{2\beta}{2\beta+1}}$, where $M$ is the sample size and $\beta$ is the Hölder smoothness of the activation function. Importantly, this rate is independent of the embedding dimension $d$, the number of tokens $N$, and the rank $r$ of the weight matrix, provided that $rd \leqslant (M/\log M)^{\frac{1}{2\beta+1}}$. These results highlight a fundamental statistical efficiency of attention-style models, even when the weight matrix and activation are not separately identifiable, and provide a theoretical understanding of attention mechanisms and guidance on training.

## 1 Introduction

The transformer architecture (Vaswani et al., 2017) has achieved remarkable success in natural language processing, computer vision, and other AI domains, with its impact most visible in large language models (LLMs) such as GPT (OpenAI, 2024), LLaMA (Touvron et al., 2023), and BERT (Devlin et al., 2019). At its core, attention mechanisms model nonlocal dependencies between input tokens through pairwise interactions, creating a function class capable of representing intricate contextual relationships.

Despite the empirical success, our theoretical understanding remains incomplete. The attention mechanism computes weighted averages of token representations using pairwise similarities, but we observe only the aggregated outputs and not the underlying interaction structure that generates them. This creates a fundamental inverse problem with critical *sample complexity* questions: can we recover the interaction function from these aggregated observations, how many samples are needed to learn token-to-token interactions for a given accuracy level, and how does the convergence rate depend on embedding dimension, number of tokens, and smoothness of the activation function? Recent phenomena like extreme attention weights on certain tokens (Sun et al., 2024; Guo et al., 2024b; Xiao et al., 2024; Wang et al., 2021) further highlight gaps in our understanding of how transformers process token interactions.

In this paper, we tackle these questions by analyzing an Interacting Particle System (IPS) model for attention-style mechanisms. Tokens are viewed as "particles," and the self-attention aggregates pairwise interactions between them. The interaction is a composite of an unknown embedding matrix

---

\*These authors contributed equally to this work.

and an unknown nonlinear activation function, both of which are learned from data. This makes the problem challenging as it is fundamentally *nonconvex*. Our IPS approach provides a natural framework for understanding how transformers process inputs with a large number of correlated tokens, moving beyond the restrictive assumption of independent, isotropic token distributions.

We summarize our main contributions below:

- We establish a connection between transformers and IPS models, enabling us to address the challenging inverse problem of inferring nonlinear interactions learned by attention mechanisms. Our analysis extends beyond the standard assumption of independent, isotropic token distributions to allow for dependent and anisotropic data.

- Inferring the interaction function is an inverse problem. We prove that under a *coercivity condition* (Lemma 3.4), this problem is well-posed in the large sample limit. This condition holds for a large class of input distributions.

- We prove that the rate of $M^{-\frac{2\beta}{2\beta+1}}$ is the optimal (up to logarithmic factors) minimax convergence rate in estimating the $2d$-dimensional pairwise interaction function, where $M$ is the sample size and $\beta$ is the Hölder exponent of the function. The error is composed of parametric and nonparametric terms. Importantly when $rd \leqslant (M/\log M)^{\frac{1}{2\beta+1}}$, the leading term is the nonparametric term $M^{-\frac{2\beta}{2\beta+1}}$, which does not depend on the embedding dimension $d$ or the rank $r$. This confirms that the attention-style model evades the curse of dimensionality.

## 1.1 RELATED WORKS

**Neural networks and IPS.** Modeling neural networks as dynamical systems through depth was introduced in Chen et al. (2018), which framed updates in ResNet architectures as the dynamics of a state vector. This perspective has been generalized to various architectures, typically treating skip connections as the evolving state across layers. Following this approach, in

Geshkovski et al. (2023; 2025)

they view tokens as interacting particles, analyze the attention as an IPS, and study clustering phenomena in continuous time (in depth).

Similarly, Dutta et al. (2021) leverages a similar framework to compute attention outputs directly from an initial state evolved over depth, thereby reducing computational costs. While these works provide valuable insights, they focus exclusively on the dynamics of tokens through the layers. To our knowledge, no existing work addresses the learning theory for estimating the pairwise interactions in such particle systems.

**Inference in attention models.** Many theoretical works have studied the learnability of attention, focusing on specific regimes. Some consider simplified variants, such as linear or random feature target attention models (Wang et al., 2020; Lu et al., 2025; Marion et al., 2025; Hron et al., 2020; Fu et al., 2023), which explore the capability of this model under simple regression tasks.

Deora et al. (2024) analyze logistic-loss optimization and prove a generalization rate under a "good" initialization.

Others consider a more specific architecture, Li et al. (2023) study the training of shallow vision transformers (ViT) and show that, with suitable initialization and enough stochastic gradient steps, a transformer with additional $\mathrm{ReLU}$ layer can achieve zero error.

Several works study softmax attention layers with trainable key and query matrices in the limit of high embedding dimension quadratically proportionate to samples with i.i.d. tokens Troiani et al. (2025); Cui et al. (2024); Cui (2025); Boncoraglio et al. (2025), which is further expanded in Troiani et al. (2025) for $\mathrm{softmax}$ attention (without the value matrix) with multiple layers.

These works have mainly focused on the linear/softmax attention model and do not consider a general interaction function. In addition, most studies assume the tokens are independent and do not draw a connection to the IPS system.

**Inference for systems of interacting particles.** There is a large body of work on the inference of systems of interacting particles; we state a few here. Parametric inference has been studied in

Amorino et al. (2023); Chen (2021); Della Maestra & Hoffmann (2023); Kasonga (1990); Liu & Qiao (2022); Sharrock et al. (2021) for the operator (drift term) and in Huang et al. (2019) for the noise variance (diffusion term). Nonparametric inference on estimating the entire operator $R_g$, but not the kernel $g$, has been studied in Della Maestra & Hoffmann (2022); Yao et al. (2022).

The closest to this study are Lu et al. (2021a; 2022; 2019); Wang et al. (2025). A key difference from these studies is that their goal is to estimate the radial interaction kernel, whereas our $2d$-dimensional pairwise interaction function is not shift-invariant due to the weight matrix. In addition, all these studies focus on IPS in general, without a clear connection to attention models.

**Activation function in transformer layer.** Recent work has shown that attention models suffer from the "extreme-token phenomenon", where certain tokens receive disproportionately high weights, creating challenges for downstream tasks (Sun et al., 2024; Guo et al., 2024b; Xiao et al., 2024; Wang et al., 2021). To address this, it was proposed to replace softmax with alternatives, such as ReLU (Guo et al., 2024a; Zhang et al., 2021), which can "turn off" irrelevant tokens, a capability that softmax lacks. While linear attention can outperform softmax in regression tasks by avoiding additional error offsets (Von Oswald et al., 2023; Katharopoulos et al., 2020; Yu et al., 2024; Han et al., 2024), it may be inferior for classification (Oymak et al., 2023). These findings suggest no universally optimal activation function exists, making the theoretical analysis of a general interaction function in transformer-type models crucial. As for vision tasks, several Vision Transformer (ViT) variants remove the softmax activation while remaining competitive. For example, Lu et al. (2021b) consider an attention mechanism based on a Gaussian kernel, and Koohpayegani & Pirsiavash (2024) apply linear attention after normalizing the Key-Query columns. Furthermore, Ramapuram et al. (2025) examine a sigmoid function as the attention activation, showing it acts as a universal function approximator and benefits from improved regularity compared to softmax attention.

**Nonparametric and Semiparametric Estimation for Neural Networks** Classical nonparametric estimation provides optimal minimax rates for simple structures. Gaïffas & Lecué (2007) provide bounds for the single index model $f(w^\top x)$ of order $M^{\frac{-2\beta}{2\beta+1}}$. For the more general projection pursuit model $f(x) = \sum_{j=1}^K f_j(\langle x, \beta_j \rangle)$, Györfi et al. (2006) shows that the minimax rate is the standard rate up to a log factor. These results directly apply to small single-layer neural networks.

Closer to deep learning, Horowitz & Mammen (2007) analyze generalized additive models with nested $k$-times differentiable compositions, showing the rate is $M^{-\frac{2k}{2k+1}}$. Schmidt-Hieber (2020) proves that connected deep ReLU networks achieve a near-optimal minimax rate (up to log factors) over a class of composed functions. In Bhattacharya et al. (2024) they study a nonparametric interaction model in high dimension settings and show sparsity assumptions and associated regularization are required in order to obtain optimal rates of convergence.

**Notation.** Throughout the paper, we use $C$ to denote universal constants independent of the sample size $M$, particles $N$ and the embedding dimension $d, r$. The notations $C_\beta$ or $C_{\beta,L}$ denote constants depending on the subscripts. We introduce the $L_\rho^2$ inner product as $\langle f, g \rangle_{L_\rho^2} = \int f(r)g(r)\rho(dr)$ and denote the $L_\rho^p$ norm by $\|f\|_{L_\rho^p}^p = \int |f(r)|^p \rho(dr)$ for all $p \geqslant 1$. For vectors $a, b \in \mathbb{R}^d$ and $A \in \mathbb{R}^{d \times d}$ we write $\langle a, b \rangle_A := a^\top A b$.

## 2 PROBLEM FORMULATION

In this section, we describe our statistical task and connect it to the attention model.

**Model setup and learning task.** We consider a model of $N$ interacting particles,

$$Y_i = \frac{1}{N-1} \sum_{j=1, j \neq i}^N \phi_\star\big(\langle X_i, X_j \rangle_{A_\star}\big) + \eta_i \tag{2.1}$$

where $\eta \in \mathbb{R}^N$ is noise as specified in Assumption 2.2, $\phi_\star : \mathbb{R} \to \mathbb{R}$ is an unknown interaction kernel, and $A_\star \in \mathbb{R}^{d \times d}$ is an unknown interaction matrix. Here, we write $\langle x, y \rangle_A := x^\top A y$ for $x, y \in \mathbb{R}^d$

and $A \in \mathbb{R}^{d \times d}$. The input $X = (X_1, \ldots, X_N)^\top \in \mathcal{C}_d^N := ([0,1]^d/\sqrt{d})^N \subset \mathbb{R}^{N \times d}$ denotes the particle positions (or token values), and the output $Y = (Y_1, \ldots, Y_N) \in \mathbb{R}^{N \times 1}$ represents the average interactions between the particles.

We observe $M$ i.i.d. samples

$$\mathcal{D}_M = \{(X^m, Y^m)\}_{m=1}^M, \qquad X^m \in \mathcal{C}_d^N := ([0,1]^d/\sqrt{d})^N, Y^m \in \mathbb{R}^N,$$

allowing the $N$ particles and their entries to be dependent.

The task is to learn the pairwise interaction function $g_\star : \mathbb{R}^d \times \mathbb{R}^d \to \mathbb{R}$,

$$g_\star(x, y) := \phi_\star\big(\langle x, y \rangle_{A_\star}\big), \qquad (x, y) \in \mathbb{R}^d \times \mathbb{R}^d, \tag{2.2}$$

from the dataset of observations $\mathcal{D}_M$.

We introduce the vectorized view of the model via the forward operator $R_g$ for any candidate interaction function $g : \mathbb{R}^d \times \mathbb{R}^d \to \mathbb{R}$ as $R_g[X]_i := \frac{1}{N-1} \sum_{j=1, j \neq i}^N g(X_i, X_j)$. Accordingly, our model in equation 2.1 becomes $Y_i = R_{g_\star}[X]_i + \eta_i$.

**Connection to self-attention layer.** We view self-attention through the lens of an IPS: tokens are "particles," and attention aggregates pairwise interactions between them. A typical self-attention layer is composed of an attention block with learnable query, key, and value matrices, $W_Q, W_K \in \mathbb{R}^{d \times d_k}$ with $d_k \leqslant d$ and $W_V \in \mathbb{R}^{d \times d_v}$ that compute

$$\text{Att}(Q, K, V) = \text{softmax}\Big(\frac{QK^\top}{\sqrt{d_k}}\Big)V, \qquad Q = XW_Q, \ K = XW_K, \ V = XW_V. \tag{2.3}$$

The attention operation is then often followed by an application of a multilayer perceptron (MLP), which maps the above into some other nonlinear function. The pairwise structure of attention motivates modeling token interactions via a scalar kernel function applied to a bilinear form of some score interaction matrix $A_\star$ that can be viewed as the learned projections through $\frac{1}{\sqrt{d_k}} W_Q W_K^\top$, i.e.,

$$\text{softmax}\Big(\frac{QK^\top}{\sqrt{d_k}}\Big) = \text{softmax}\Big(X A_\star X^\top\Big), \quad X A_\star X^\top = \big(X_i^\top \tfrac{W_Q W_K^\top}{\sqrt{d_k}} X_j\big)_{1 \leqslant i, j \leqslant N}.$$

The interaction function in equation 2.2 can be interpreted as either a function induced by the MLP and the softmax function, or as a general activation function with a constant value matrix; see more details in Appendix A.

As stated in the related work, such a setup for a general activation function is often desirable due to the extreme-token phenomenon (Sun et al., 2024; Guo et al., 2024b; Xiao et al., 2024; Wang et al., 2021)

Consequently, the problem of estimating $g_\star$

from the samples described in equation 2.1 is analogous to the joint estimation of the activation function and weight matrix governing nonlocal token–token interactions in a single-layer self-attention mechanism.

**Goal of this study.** Our goal is to characterize the optimal (minimax) convergence rate of estimators of $g_\star$ as the sample size $M$ grows.

To assess the estimation error for the interaction function, we introduce empirical measures over pairs of particles/tokens $(x, y)$. Termed *exploration measures*, they quantify the extent to which the data explores the argument space relevant to the function.

**Definition 2.1 (Exploration measure)** *Let $\{X^m \in \mathcal{C}_d\}_{m=1}^M$ be sampled sequence. Define the* empirical exploration measure *of off-diagonal pairs of particles*

$$\rho_M(B) := \frac{1}{MN(N-1)} \sum_{m=1}^M \sum_{i=1}^N \sum_{j=1, j \neq i}^N \mathbf{1}_{\{(X_i^m, X_j^m) \in B\}}$$

*and the* population exploration measure *as $\rho(B) := \lim_{M \to \infty} \rho_M(B) = \mathbb{E}[\rho_M(B)]$,*

*for any Lebesgue measurable set $B \subset \mathbb{R}^d \times \mathbb{R}^d$.*

We aim to provide matching upper and lower bound rates for the $L^2_\rho$ error of the estimator $\widehat{g}$, so as to obtain a minimax convergence rate:

$$\mathbb{E}\Big[\|\widehat{g} - g_\star\|^2_{L^2_\rho}\Big] \approx M^{-\frac{2\beta}{2\beta+1}}, \quad \text{as } M \to \infty, \tag{2.4}$$

where $\beta$ is the Hölder exponent of $g_\star$ (which is determined by the smoothness of $\phi_\star$). This then demonstrates that the attention model is not susceptible to the curse of dimensionality. In particular, we aim to characterize the dependence of the rate on the embedding dimension $d$, the rank $r$ of the interaction matrix, and the number of tokens $N$.

### 2.1 Assumptions on the data distribution

We now state the assumptions on the distributions of the input and the noise used throughout this work. We do not assume that the $N$ tokens are independent of each other.

**Assumption 2.1 (Data Distribution)** *We assume the entries of the $\mathcal{C}^N_d = ([0,1]^d/\sqrt{d})^N$-valued random variable $X = (X_1, \ldots, X_N)$ satisfy the following conditions:*

(A1) *The components of the random vector $X = (X_1, \ldots, X_N)$ are exchangeable.*

(A2) *For each $i \in \{1, \ldots, N\}$ and any $j \neq j'$ with $j, j' \neq i$, there exists a $\sigma$-algebra $\mathcal{X}_i \supseteq \sigma(X_i)$ such that $X_j$ and $X_{j'}$ are conditionally independent given $\mathcal{X}_i$.*

(A3) *The joint distribution of $(X_i, X_j)$ has a continuous density function for each pair.*

These assumptions simplify the inverse problem and may be replaced by weaker constraints; see Wang et al. (2025) for a discussion and references therein. The exchangeability in (A1) simplifies the exploration measure in Lemma B.1. It enables the coercivity condition for the inverse problem to be well-posed, as detailed in Lemma 3.4, and is only used in the upper bound in Theorem 3.1.

The continuity in Assumption (A3) ensures that the exploration measure has a continuous density, which is used in proving the lower minimax rate Theorem 4.4.

We next specify the noise setting. Assumption 2.2 details the constraints we assume for the noise:

**Assumption 2.2 (Noise Distribution)** *The noise $\eta \in \mathbb{R}^N$ is centered and independent of the random array $X$. Moreover, we assume the following conditions:*

(B1) *The entries of the noise vector $\eta = (\eta_1, \ldots, \eta_N)$ are sub-Gaussian in the sense that for all $i$, $\mathbb{E}[e^{c\eta_i^2}] < \infty$ for some $c > 0$.*

(B2) *There exists a constant $c_\eta > 0$ such that The density $p_\eta$ of $\eta$ satisfies the following:*

$$\int_{\mathbb{R}^N} p_\eta(u) \log \frac{p_\eta(u)}{p_\eta(u+v)} du \leqslant c_\eta \|v\|^2, \quad \forall v \in \mathbb{R}^N. \tag{2.5}$$

We note that assumptions (B1) and (B2) hold for instance for Gaussian noise $\eta \sim \mathcal{N}(0, \sigma^2_\eta I_N)$ with $c_\eta = 1/(2\sigma^2_\eta)$.

### 2.2 Function classes and model/estimator assumptions

We introduce the functional classes where $g_\star$ lies. Our goal is to consider as large a class of functions as possible while also tracking the properties of the models $\phi_\star$ that control the rate. For that purpose, we introduce the Hölder class and assume that $\phi_\star$ satisfies some smoothness order of $\beta$.

**Definition 2.2 (Hölder classes)** *For $\beta, L, \bar{a} > 0$, the Hölder class $\mathcal{C}^\beta(L, \bar{a})$ on $[-\bar{a}, \bar{a}]$ is given by*

$$\mathcal{C}^\beta(L, \bar{a}) = \Big\{ f : [-\bar{a}, \bar{a}] \to \mathbb{R} : |f^{(l)}(x) - f^{(l)}(y)| \leqslant L|x - y|^{\beta-l}, \forall x, y \in [-\bar{a}, \bar{a}] \Big\}, \tag{2.6}$$

*where $f^{(j)}$ denotes the $j$-th order derivative of functions $f$ and $l = \lfloor \beta \rfloor$.*

Low-rank Key and Query matrices often play an important role in the attention model. To keep track of the effects of the rank on the minimax rate, we introduce the following matrix class for the interaction matrix $A_\star$, which is the product of the Key and Query matrices.

**Definition 2.3 (Interaction matrix class)** *For $\bar{a} > 0$, the $d$-dimensional matrix class $\mathcal{A}_d(r, \bar{a})$ with rank $r \in \mathbb{N}$ and $2 \leqslant r \leqslant d$ is given by*

$$\mathcal{A}_d(r, \bar{a}) = \{A \in \mathbb{R}^{d \times d} : 2 \leqslant \text{rank}(A) \leqslant r, \ \|A\|_{\text{op}} \leqslant \bar{a}\}. \tag{2.7}$$

Combining both classes, we consider the following function class $\mathcal{G}_r^\beta$ for all the possible pair-wise interaction functions.

**Definition 2.4 (Target function class)** *Given $L, B_\phi, \bar{a} > 0$ and rank $r \geqslant 2$, $\beta > 0$ define*

$$\mathcal{G}_r^\beta(L, B_\phi, \bar{a}) = \left\{g_{\phi, A}(x, y) := \phi(x^\top A y) \ : \phi \in \mathcal{C}^\beta(L, \bar{a}), \ \|\phi\|_\infty \leqslant B_\phi, A \in \mathcal{A}_d(r, \bar{a})\right\}. \tag{2.8}$$

*Moreover, for any $g \in \mathcal{G}_r^\beta = \mathcal{G}_r^\beta(L, B_\phi, \bar{a})$, it follows that $|R_g[X]_i| \leqslant B_\phi$. For technical reasons, we require $L \leqslant B_\phi(2\bar{a})^\beta$. This holds without loss of generality for any $\bar{a} \geqslant 1$ and $L \leqslant B_\phi$.*

We provide both lower and upper bounds for the possible error rate by the number of samples for the interaction $g(\cdot, \cdot) \in \mathcal{G}_r^\beta$. We consider the following functional class for our estimator:

**Definition 2.5 (Estimator function class)** *Let $s := \max(\lfloor \beta \rfloor, 1)$ and $K_M \in \mathbb{N}$. Let $\Phi_{K_M}^s$ denote the class of piecewise polynomials of degree $s$, defined on $K_M$ equal sub-intervals of $[-\bar{a}, \bar{a}]$.*

*The corresponding* estimator *model class is*

$$\mathcal{G}_{r, K_M}^s := \left\{g_{\phi, A} : \ \phi \in \Phi_{K_M}^s, \ \|\phi\|_\infty + \|\phi'\|_\infty \leqslant B_\phi, \ A \in \mathcal{A}_d(r, \bar{a})\right\} \subseteq \mathcal{G}_r^\beta. \tag{2.9}$$

## 3 UPPER BOUND

In this section, we provide an upper bound on estimating the token-token interaction. We propose the following estimator $\widehat{g}_M(x, y) = \hat{\phi}(\langle x, y \rangle_{\hat{A}})$

as the empirical risk minimizer over the functional class 2.5

$$\begin{cases} \widehat{g}_M = \underset{g_{\phi, A} \in \mathcal{G}_{r, K_M}^s}{\arg \min} \ \mathcal{E}_M(g_{\phi, A}) := \frac{1}{N} \sum_{i=1}^N \mathcal{E}_M^{(i)}(g_{\phi, A}) \quad \text{with} \\ \mathcal{E}_M^{(i)}(g_{\phi, A}) := \frac{1}{M} \sum_{m=1}^M \|Y_i^m - R_{g_{\phi, A}}[X^m]_i\|^2. \end{cases} \tag{3.1}$$

Here, $R_{g_{\phi, A}}[X]_i = \frac{1}{N-1} \sum_{j=1, j \neq i}^N g_{\phi, A}(X_i, X_j)$ is the forward operator with interaction function $g_{\phi, A}$. Our goal is to prove that the estimator $\widehat{g}_M$ achieves the optimal upper bound. The large sample limit of $\mathcal{E}_M(g_{\phi, A})$ is then

$$\mathcal{E}_\infty(g_{\phi, A}) := \lim_{M \to \infty} \mathcal{E}_M(g_{\phi, A}) = \frac{1}{N} \mathbb{E}\left[\|Y - R_{g_{\phi, A}}[X]\|_2^2\right].$$

The $i$-th error $\mathcal{E}_\infty^{(i)}(g_{\phi, A})$ for any $1 \leqslant i \leqslant N$ is defined in the same manner.

The next theorem states that this estimator achieves the nearly optimal rate in estimating the interaction function. This rate matches the lower bound in Theorem 4.4 up to a logarithmic factor. Its proof is deferred to Appendix B.1.

**Theorem 3.1** *Suppose $rd \leqslant (M/\log M)^{\frac{1}{2\beta+1}}$. Consider the estimator $\widehat{g}_M$ defined in equation 3.1 computed on data $M$ i.i.d. observations satisfying Assumptions 2.1 and (B1). Then, for $\widehat{g}_M$ defined in equation 3.1 it holds that*

$$\limsup_{M \to \infty} \sup_{g_\star \in \mathcal{G}_r^\beta(L, B_\phi, \bar{a})} \mathbb{E}\left[M^{\frac{2\beta}{2\beta+1}} \|\widehat{g}_M - g_\star\|_{L_\rho^2}^2\right] \lesssim C_{N, L, \bar{a}, \beta, s}, \tag{3.2}$$

*where $C_{N, L, \bar{a}, \beta, s} = N[C_1^\beta \frac{L^2(s\bar{a})^{2\beta}}{(s!)^2} + C_2]$ for some universal positive constants $C_1, C_2$.*

**Remark 3.2** *The symbol $\lesssim$ indicates that the upper bound holds up to a logarithmic factor of $(\log M)^{\frac{2\beta}{2\beta+1}+4\max(2\beta,1)}$. We believe this factor can be improved, as it currently creates a gap between our upper and lower bounds, representing a limitation of our methods. It is worth noting that by working with uniformly bounded noise, this factor can be simplified (e.g., see Theorem 22.2 in Györfi et al. (2006)). In simpler settings, such as standard regression or when the interaction matrix $A$ is constant (e.g., for Euclidean distances), this logarithmic factor can be removed using more advanced techniques. This topic is discussed in several works, including Wang et al. (2025); Györfi et al. (2006); Van der Vaart (2000) and the references therein. However, in our model, the optimization depends on both the interaction matrix $A$ and the function $\phi$, which makes the problem non-convex. This difficulty makes the aforementioned techniques harder to implement. We therefore leave this for future work.*

**Remark 3.3** *This theorem demonstrates that the nonparametric component of the estimation error of $g_\star$ avoids the usual curse of dimensionality: the leading term $M^{-\frac{2\beta}{2\beta+1}}$ does not depend on $d$. The dimension enters only through the additional parametric contribution associated with estimating $A_\star$, which is dominated under $rd \leqslant (M/\log M)^{\frac{1}{2\beta+1}}$.*

The proof extends the technique in Györfi et al. (2006, Theorem 22.2) originally developed for the projection pursuit algorithm for multi-index models. Our setup differs from the multi-index setup in which one estimates $Y = \sum_{i=1}^{K} f_i(b_i^\top X) + \eta$ with $\{f_i : \mathbb{R} \to \mathbb{R} \text{ and } b_i \in \mathbb{R}^d\}_{i=1}^K$ from sample data $\{(X^m, Y^m)\}_{m=1}^M$, where the data $Y$ depends locally on projected values of single particle $X$. Here, the attention-style model involves averaging multiple values of the pairwise interaction function, which is a composition of the unknown $\phi_\star$ and $A$. This nonlocal dependence, combined with the mixture of parametric and nonparametric estimations, presents a significant challenge.

We list below the main challenges we address in the proof of Theorem 3.1.

1. *Nonlocal dependency.* The nonlocal dependence presents a challenge in estimating the interaction function. The forward operator $R_g[X]$ depends on the $g$ non-locally through the weighted sum of multiple values of $g$ of pairwise interaction. Thus, this is a type of inverse problem that raises significant hurdles in both well-posedness and the construction of estimators to achieve the minimax rate. To address these challenges, we show first that the inverse problem in the large sample limit is well-posed for a large class of distributions of $X$ satisfying Assumption 2.1. A crucial condition for well-posedness of this inverse problem is the *coercivity condition* studied in Li & Lu (2023); Li et al. (2021); Lu et al. (2019); Wang et al. (2025):

$$\frac{1}{N-1}\mathbb{E}\left[\|\widehat{g}_M - g_\star\|_{L^2_\rho}^2\right] \leqslant \mathcal{E}_\infty(\widehat{g}_M) - \mathcal{E}_\infty(g_\star).$$

   We prove this condition holds for a general function in our class in Lemma 3.4. Importantly, differing from these studies where the goal is to estimate the radial interaction kernel, our interaction is not shift-invariant due to the matrix and it is a $2d$-dimensional pairwise interaction function.

2. *Tail decay noise distribution.* The proof in Györfi et al. (2006) is limited to bounded noise. We provide a more general statement for any sub-Gaussian noise. This is done by decomposing the error bound now into three parts.

$$\mathcal{E}_\infty(\widehat{g}_M) - \mathcal{E}_\infty(g_\star) \leqslant \mathbb{E}[T_{1,M}] + \mathbb{E}[T_{2,M}] + \mathbb{E}[T_{3,M}]. \tag{3.3}$$

   The first two terms are a clever form of a bias-variance decomposition applied to a truncated version of the target. To bound these terms, we use a similar technique as in Györfi et al. (2006). To control the last term $T_{3,M}$ due to the truncation, we apply a lemma proved in (Kohler & Mehnert, 2011, Lemma 2).

3. *Covering numbers estimates.* Since our interaction is of the form $\frac{1}{N-1}\sum_{j=1}^N \phi(X_i^\top A X_j)$ instead of working in the space of vectors, we provide a covering estimate for the class of matrices with rank less than or equal to $r$. This is done in Lemma B.3.

The next lemma proves the crucial condition for the well-posedness of the inverse problem of estimating the interaction function. This Lemma assumes exchangeability and allows us to extract the error of the mean interaction. Its proof is based on the exchangeability of the particle distribution and is postponed to Appendix B.1.

**Lemma 3.4 (Coercivity)** *Let* $g, g_\star \in \mathcal{G}_r^\beta(L, B_\phi, \bar{a})$. *Under exchangeability of* $(X_i)_{i=1}^N$ *in Assumption* (A1) *, we have*

$$\frac{1}{N-1} \|g - g_\star\|_{L_\rho^2}^2 \leqslant \mathcal{E}_\infty(g) - \mathcal{E}_\infty(g_\star).$$

## 4 LOWER BOUND

This section establishes a lower bound for estimating $g_\star(x, y) := \phi_\star(x^\top A_\star y)$ that matches the upper bound in Theorem 3.1; together, these results determine the minimax rate.

The main challenge lies in the nonlocal dependence of the output $Y_i$ on $g_\star$, which is determined through averaging over all particles, as we don't directly observe any value of $g_\star$. Thus, the estimation of $g_\star$ is a deconvolution-type inverse problem, which is harder than estimating the single index model $Y = f(b^\top X) + \eta$ in Gaïffas & Lecué (2007). Importantly, the nonlinear joint dependence of $g_\star$ on the unknown $\phi_\star$ and $A_\star$ further complicates the problem.

We address the challenge by first reducing the supremum over all $g_\star$ to the supremum over all $\phi_\star$ with a fixed $A_\star \in \mathcal{A}_d(r, \bar{a})$, building on a technical result in Lemma 4.1. This reduces the problem to the minimax lower bound of estimating the interaction kernel $\phi_\star$ only. We derive this lower bound using the scheme in Wang et al. (2025), a variant of the Fano-Tsybakov method in Tsybakov (2008).

Let $A_\star \in \mathcal{A}_d(r, \bar{a})$ and let

$$U_{ij} := X_i^\top A_\star X_j, \quad U \sim p_U(u) := \frac{1}{N(N-1)} \sum_{i=1}^N \sum_{j=1, j \neq i}^N p_{U_{ij}}(u), \tag{4.1}$$

where $p_{U_{ij}}$ denotes the probability density of $U_{ij}$. Here, the density $p_{U_{ij}}$ exists and is continuous because $\text{rank}(A_\star) \geqslant 2$ and the joint density of $(X_i, X_j)$ exists by Assumption (A3), see Lemma C.1. Hence, the density $p_U$ is continuous. Furthermore, since $\|A_\star\|_{\text{op}} \leqslant \bar{a}$ and $X_i \in \mathcal{C}_d$, we have $|U_{ij}| \leqslant \bar{a}$ and $\text{supp}(p_U) \subset [-\bar{a}, \bar{a}]$. In particular, when the distribution $X$ is exchangeable, we have $p_{U_{ij}}(u) = p_{U_{12}}(u) = p_U(u)$ for all $(i, j)$, $u \in [-\bar{a}, \bar{a}]$. However, our proof below works for non-exchangeable distributions.

The next lemma allows us to reduce the supremum over all $g_\star(x, y) = \phi_\star(x^\top A_\star y)$ to all $\phi_\star$ by bounding $\|\hat{g} - g_\star\|_{L_\rho^2}^2$ from below by $\|\hat{\psi} - \phi_\star\|_{L_{p_U}^2}^2$ for a function $\hat{\psi}$ determined by $\hat{g}$ and $A_\star$. Its proof can be found in Section C.

**Lemma 4.1** *Suppose Assumption* (A3) *holds. Let* $A_\star, \hat{A} \in \mathcal{A}_d(r, \bar{a})$. *Recall the definitions of* $U_{ij}$ *and* $U \sim p_U$ *(defined according to* $A_\star$*) in equation 4.1. Let* $\phi_\star, \hat{\phi} \in L_{p_U}^2$, *and define a function* $\hat{\psi}$ *that is determined by* $(\hat{\phi}, \hat{A}, A_\star)$ *and the distribution of* $X$ *as*

$$\hat{\psi}_{ij}(u) := \mathbb{E}\big[\hat{\phi}(X_i^\top \hat{A} X_j)\big|U_{ij} = u\big], \quad \hat{\psi}(u) := \sum_{i=1}^N \sum_{j=1, j \neq i}^N \frac{p_{U_{ij}}(u)}{N(N-1)p_U(u)} \hat{\psi}_{ij(u)}. \tag{4.2}$$

*Then, the following inequality holds:*

$$\|\hat{g} - g_\star\|_{L_\rho^2}^2 \geqslant \int_{-\bar{a}}^{\bar{a}} \big|\hat{\psi}(u) - \phi_\star(u)\big|^2 p_U(u)\mathrm{d}u.$$

The next lemma constructs a finite family of hypothesis functions that are well-separated in $L_{p_U}^2$, while their induced distributions remain close with a slowly increasing total Kullback-Leibler divergence, enabling the application of Fano's method to derive the minimax lower bound. Its proof follows the scheme in Wang et al. (2025) and is postponed to Section C.

**Lemma 4.2** *For each data set* $\{(X^m, Y^m)\}_{m=1}^M$ *sampled from the model* $Y = R_{\phi_\star, A_\star}[X] + \eta$, *where* $A_\star \in \mathcal{A}_d(r, \bar{a})$ *satisfying assumptions* (B2) *and* (A3), *there exists a set of hypothesis functions* $\{\phi_{0,M} \equiv 0, \phi_{1,M}, \cdots, \phi_{K,M}\}$ *and positive constants* $\{C_0, C_1\}$ *independent of* $M, N, d, r$, *where*

$$K \geqslant 2^{\bar{K}/8}, \quad \text{with } \bar{K} = \lceil c_{0,N} M^{\frac{1}{2\beta+1}} \rceil, \quad c_{0,N} = C_0 N^{\frac{1}{2\beta+1}}, \tag{4.3}$$

*such that the following conditions hold:*

(D1) *Holder continuity:* $\phi_{k,M} \in \mathcal{C}^\beta(L,\bar{a})$ *and* $\|\phi_{k,M}\|_\infty \leqslant B_\phi$ *for each* $k = 1, \cdots, K$;

(D2) $2s_{N,M}$*-separated:* $\|\phi_{k,M} - \phi_{k',M}\|_{L^2_{p_U}} \geqslant 2s_{N,M}$ *with* $s_{N,M} = C_1 c_{0,N}^{-\beta} M^{-\frac{\beta}{2\beta+1}}$;

(D3) *Kullback-Leibler divergence estimate:* $\frac{1}{K} \sum_{k=1}^K D_{\mathrm{KL}}(\bar{\mathbb{P}}_k, \bar{\mathbb{P}}_0) \leqslant \alpha \log(K)$ *with* $\alpha < 1/8$,

*where* $\bar{\mathbb{P}}_k(\cdot) = \mathbb{P}_{\phi_{k,M}}(\cdot \mid X^1, \ldots, X^M)$ *and* $p_U$ *is the density of* $U$ *defined in equation 4.1.*

The following theorem provides a lower minimax rate for estimating $\phi_\star$ when $A_\star$ is given. Its proof is available in Section C.

**Theorem 4.3** *Suppose Assumptions* (A3) *and* (B2) *hold. Let* $p_U$ *be the density of* $U$ *defined in equation 4.1. Then, for any* $\beta > 0$*, there exists a constant* $c_0 > 0$ *independent of* $M$, $d$, $r$ *and* $N$ *such that*

$$\liminf_{M \to \infty} \inf_{\hat{\psi}_M \in L^2_{p_U}} \sup_{\substack{\phi_\star \in \mathcal{C}^\beta(L,\bar{a}) \\ \|\phi_\star\|_\infty \leqslant B_\phi}} \mathbb{E}_{\phi_\star}\left[ M^{\frac{2\beta}{2\beta+1}} \|\hat{\psi}_M - \phi_\star\|^2_{L^2_{p_U}} \right] \geqslant c_0 N^{-\frac{2\beta}{2\beta+1}} \tag{4.4}$$

*where* $\hat{\psi}_M$ *is estimated based on the observation model with* $M$ *i.i.d. samples.*

Following the above results, we can now provide a lower bound for the convergence rate when estimating $g_\star$ over all possible estimators in the worst-case scenario.

**Theorem 4.4 (Minimax lower bound)** *Suppose Assumptions* (A3) *and* (B2) *hold. Then, for any* $\beta > 0$ *there exists a constant* $c_0 > 0$ *independent of* $M$, $d$, $r$ *and* $N$*, such that the following inequality holds:*

$$\liminf_{M \to \infty} \inf_{\hat{g}} \sup_{g_\star \in \mathcal{G}^\beta_r(L,B_\phi,\bar{a})} M^{\frac{2\beta}{2\beta+1}} \mathbb{E}\left[ \|\hat{g} - g_\star\|^2_{L^2_\rho} \right] \geqslant c_0 N^{-\frac{2\beta}{(2\beta+1)}} \tag{4.5}$$

*where the infimum* $\inf_{\hat{g}}$ *is taken over all* $\hat{g}(x,y) = \hat{\phi}(x^\top \hat{A} y)$ *with* $\hat{A} \in \mathcal{A}_d(r,\bar{a})$ *and* $\hat{\phi}$ *such that* $\hat{g} \in L^2_\rho$.

## 5 NUMERICAL SIMULATIONS

In this section, we empirically verify the convergence rates predicted by our theory, emphasizing their independence from the ambient dimension $d$ and their dependence on the activation function's smoothness.

For all experiments, we use B-splines to represent the ground-truth activation $\phi_\star$: a degree-$p$ B-spline is $C^{p-1}$, so the degree directly controls the smoothness (Lyche et al., 2017). B-splines are linear in their basis coefficients, allowing us to efficiently compute an optimal coefficient estimator by least squares. Our estimator for the interaction function $\hat{g}$ exploits this structure: we first fit $\phi_\star$ in the B-spline basis by least squares, then approximate the fitted activation with a multi-layer perceptron to enable backpropagation when estimating $A_\star$. This design enables us to control both the smoothness and the approximation accuracy of $\hat{g}$, ensuring that it achieves the minimax rate. Full simulation and parameter details appear in Appendix D.

Our experiments confirm the theoretical minimax rates.

- *Independence from the ambient dimension d.* Figure 1(a) compares convergence across embedding dimensions $d \in \{1, 5, 30\}$. In the log-log plots, the slopes (which encode the rates) are nearly parallel and close to the theoretical exponent $-2\beta/(2\beta + 1)$ for all three dimensions, indicating that the convergence rate is independent of $d$.

- *Dependence on the activation function's smoothness.* Figure 1(b) reports rates for varying smoothness exponents $\beta$, controlled by the B-spline degree used to represent $\phi_\star$. As the spline degree (and hence $\beta$) increases, the log-log slope steepens as predicted by theory: for example, the empirical slopes are $\approx -0.81$ for degree $P = 3$ and $\approx -0.899$ for $P = 8$, closely matching the theoretical values $-0.80$ and $-0.933$.

The two plots illustrate that the minimax rate is fully determined by the smoothness $\beta$ and it is doesn't change with the dimension.

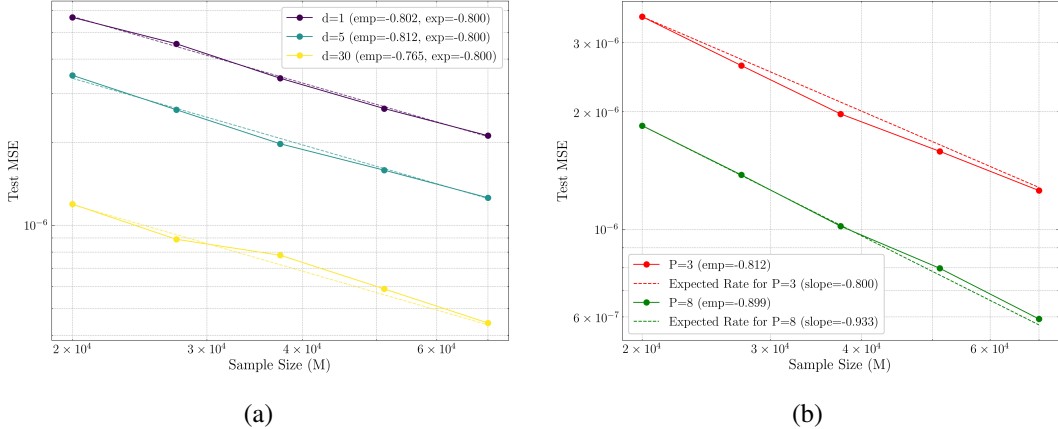

(a)                 (b)

Figure 1: **(a)** Convergence rates with $d \in \{1, 5, 30\}$. Composed test Mean Squared Error (MSE) vs. sample size $M$ in log scale; dashed lines show the expected rate $M^{-2\beta/(2\beta+1)}$; and the markers represent the median across seeds. The convergence rates are nearly the same for different values of $d$. **(b)** Convergence rates with varying smoothness exponents, which are controlled by the spline degree of $\phi_\star$ and the estimator, with $P_{\text{true}} = P_{\text{est}} \in \{3, 8\}$, corresponding to $\beta \in \{2, 7\}$ and expected slopes $-0.800$ and $-0.933$. The parameters in each simulation are described in Appendix D.

## 6 CONCLUSIONS

We have established minimax convergence rates for estimating the pairwise interaction functions in self-attention style models. The rates include a leading term $M^{-2\beta/(2\beta+1)}$ that is independent of $d$ which controls the parametric rate of estimating $A_\star$ when the sample size adheres to $rd \leqslant (M/\log M)^{\frac{1}{2\beta+1}}$ condition. This result is achieved using a direct connection to interacting particle systems (IPS), we have proved that under a coercivity condition, one can learn the interaction function at an optimal rate $M^{-2\beta/(2\beta+1)}$ with $\beta$ being the smoothness of the function. Notably, under the $rd \leqslant (M/\log M)^{\frac{1}{2\beta+1}}$ condition, this rate is independent of both the embedding dimension and the number of tokens. Our analysis extends beyond the standard assumption of independent, isotropic token distributions to allow for correlated and anisotropic token distributions. These results illuminate how attention can avoid the curse of dimensionality in high-dimensional regimes. Viewing attention through the IPS lens suggests a broad research agenda for understanding the attention models. Promising next steps include extending the theory to multi-head attention, residual connections and self-attention interactions induced by the value matrix. Advances in these directions will improve our understanding of learning mechanisms and generalization in transformers.

## ACKNOWLEDGMENTS

This project is part of the NSF-BSF grant 0603624011 and DMS 2511283. XW was partially supported by the Sun Yat-sen University research startup. IS was partially supported by the Israel Science Foundation grant 777/25 and by the Alon fellowship. FL was partially supported by the NSF grant DMS 2238486.

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

## A  APPENDIX: REDUCTION FROM ATTENTION TO IPS ATTENTION MODEL

In this section, we provide a direct connection between the IPS attention model and the softmax self-attention layer, which typically includes an additional normalization step. Consider a sequence of tokens $\{X_i\}_{i=1}^N$. The output of the softmax self-attention layer is typically composed of an attention block with learnable query, key, and value matrices, $W_Q, W_K \in \mathbb{R}^{d \times d_k}$ with $d_k \leqslant d$ and $W_V \in \mathbb{R}^{d \times d_v}$ that compute

$$Y = \text{Att}(Q, K, V) = \text{softmax}\left(\frac{QK^\top}{\sqrt{d_k}}\right)V, \qquad Q = XW_Q, \ K = XW_K, \ V = XW_V. \quad (\text{A.1})$$

As explained in the main text, we denote by $A = \frac{1}{\sqrt{d_k}} W_Q W_K^\top$ the score interaction matrix. Using the definition of the softmax function, the output of the softmax self-attention layer for each particle can then be written

$$Y_i = \sum_{j=1}^N \frac{e^{\beta X_i^\top A X_j}}{Z_i[X]} V_j, \quad Z_i[X] = \sum_{\ell=1}^N e^{\beta X_i^\top A X_\ell}$$

with $\beta > 0$ being the inverse temperature parameter. When the number of particles is large, the partition function $Z_i[X]$ concentrates around its mean-field value with respect to the empirical distribution of the particles. If we denote by $\mu$ the continuum limit of the empirical measure, then $Z_i[X] \approx N\mathcal{Z}_i = N \int e^{\beta X_i^\top A y} \mathrm{d}\mu(y)$ conditioned on the $i$-th particle.

For the IPS surrogate we consider in this paper, we adopt two standard simplifications (Sander et al., 2022; Geshkovski et al., 2025; Bruno et al., 2025): we set $d_v = 1$, and treat $\mathcal{Z}_i$ as a constant (independent of $X$) that can be absorbed into the nonlinearity, and focus only on the self-interaction for $i \neq j$, and setting $V_j$ to be a constant, we get our IPS Attention Model:

$$Y_i = \frac{1}{N} \sum_{j \neq i} \phi(X_i^\top A X_j).$$

This reduction is similar in spirit to the surrogate model (USA) presented in Geshkovski et al. (2025). We note that a possible extension of our model to account for the softmax normalization would be to learn a function for each particle, $\phi_i$. We suspect it will not change the overall rate. In fact, as stated in Geshkovski et al. (2025), this reduction seems to capture the essence of the dynamics of the self-attention (SA) model. Therefore, to simplify the setting, we focus on estimating a single function.

## B  APPENDIX: UPPER BOUND PROOFS

We begin by reducing the distribution of the pair-wise particles to the distribution of one pair by exchangeability. The exchangeability not only simplifies the proof of the upper bound, but also provides a sufficient condition for the coercivity, which makes the inverse problem well-posed.

**Lemma B.1 (Exploration measure under exchangeability)** *Under Assumption 2.1, the measure $\rho$ is the distribution of $(X_1, X_2) \in \mathbb{R}^d \times \mathbb{R}^d$ and has a continuous density.*

**Proof.** The exchangeability in Assumption 2.1 implies that the distributions of $(X_i, X_j)$ and $(X_1, X_2)$ are the same for any $i \neq j$. Hence, by definition, the exploration measure is the distribution of the random variables $(X_1, X_2)$:

$$\rho(B) = \mathbb{P}((X_1, X_2) \in B)$$

which has a continuous density by Assumption 2.1. □

### B.1  PROOF OF THE UPPER BOUND IN THEOREM 3.1

In this section, we provide the proof of the upper bound.

We begin with the proof of the key coercivity lemma, which is crucial in bounding the error of the interaction function and making the inverse problem well-posed in the large sample limit.

**Proof of Lemma 3.4** Recall $R_g(X)_i = \frac{1}{N-1} \sum_{j \neq i} g(X_i, X_j)$. By definition

$$\mathcal{E}_\infty(g) - \mathcal{E}_\infty(g_\star) = \frac{1}{N} \mathbb{E}\langle R_{g-g_\star}[X], R_{g-g_\star}[X]\rangle = \frac{1}{N(N-1)^2} \sum_{i=1}^N \sum_{j \neq i} \sum_{j' \neq i} \mathbb{E}\langle \Delta_{ij}, \Delta_{ij'}\rangle,$$

where $\Delta_{ij} = (g - g_\star)(X_i, X_j)$ and $\sum_{j \neq i} = \sum_{j=1, j \neq i}^N$. By exchangeability,

$$\frac{1}{N(N-1)^2} \sum_{i=1}^N \sum_{j \neq i} \sum_{j' \neq i} \mathbb{E}\langle \Delta_{ij}, \Delta_{ij'}\rangle = \frac{1}{N-1} \mathbb{E}\|\Delta_{12}\|^2 + \frac{N-2}{N-1} \mathbb{E}\langle \Delta_{12}, \mathbb{E}[\Delta_{13} \mid X_1]\rangle$$

$$\geqslant \frac{1}{N-1} \mathbb{E}\|\Delta_{12}\|^2,$$

since $\mathbb{E}\|\mathbb{E}[\Delta_{13} \mid X_1]\|^2 \geqslant 0$. The statement of the Lemma follows. □

**Proof of Theorem 3.1** The proof is divided into five steps.

**Step 1: Error decomposition.** In this step, we decompose the mean squared error $\mathbb{E}[\|\hat{g}_M - g_\star\|_{L_\rho^2}^2]$ to two terms. Using Lemma 3.4, i.e., the coercivity condition and the definition of $\mathcal{E}_\infty(g) = $

$\frac{1}{N}\mathbb{E}\left[\|Y - R_g[X]\|^2\right]$, we have for $c_{\mathcal{H}} = \frac{1}{N-1}$

$$c_{\mathcal{H}}\mathbb{E}\left[\int |\widehat{g}_M(x,y) - g_\star(x,y)|^2 \mathrm{d}\rho(x,y)\right]$$

$$\leqslant \mathcal{E}_\infty(\widehat{g}_M) - \mathcal{E}_\infty(g_\star)$$

$$= \frac{1}{N}\mathbb{E}[\|Y - R_{\widehat{g}_M}[X]\|^2] - \frac{1}{N}\mathbb{E}\left[\|Y - R_{g_\star}[X]\|^2\right]$$

$$= \frac{1}{N}\mathbb{E}\left[\mathbb{E}\left[\|Y - R_{\widehat{g}_M}[X]\|^2 \mid \mathcal{D}_M\right]\right] - \frac{1}{N}\mathbb{E}\left[\|Y - R_{g_\star}[X]\|^2\right]. \tag{B.1}$$

Let $B_M := c_1 \log(M)$ with some constant $c_1 > 0$ and $Y_M := \min(B_M, \max(-B_M, Y))$. Let us denote

$$T_{1,M} := 2[\mathcal{E}_M(\widehat{g}_M) - \mathcal{E}_M(g_\star)] \tag{B.2}$$

and

$$T_{2,M} := \frac{1}{N}\mathbb{E}\left[\|Y_M - R_{\widehat{g}_M}[X]\|^2 \mid \mathcal{D}_M\right] - \frac{1}{N}\mathbb{E}\left[\|Y_M - R_{g_\star}[X]\|^2\right] - T_{1,M}, \tag{B.3}$$

$$T_{3,M} := \frac{1}{N}\mathbb{E}\left[\|Y - R_{\widehat{g}_M}[X]\|^2 \mid \mathcal{D}_M\right] - \frac{1}{N}\mathbb{E}\left[\|Y - R_{g_\star}[X]\|^2\right] \tag{B.4}$$

$$- \frac{1}{N}\mathbb{E}\left[\|Y_M - R_{\widehat{g}_M}[X]\|^2 \mid \mathcal{D}_M\right] + \frac{1}{N}\mathbb{E}\left[\|Y_M - R_{g_\star}[X]\|^2\right].$$

By equation B.1, we can decompose the upper bound of the mean squared error as

$$\mathbb{E}\left[\|\widehat{g}_M - g_\star\|_{L^2_\rho}^2\right] = \mathbb{E}\left[\int |\widehat{g}_M(x,y) - g_\star(x,y)|^2 \mathrm{d}\rho(x,y)\right]$$

$$\leqslant c_{\mathcal{H}}^{-1}\left(\mathbb{E}[T_{1,M}] + \mathbb{E}[T_{2,M}] + \mathbb{E}[T_{3,M}]\right). \tag{B.5}$$

We shall proceed with our proof by bounding $\{\mathbb{E}[T_{i,M}]\}_{i=1}^3$ in the following Steps 2-4 via approximation error estimate, covering number estimate, and sub-Gaussian property, respectively.

**Step 2: Bounding $\mathbb{E}[T_{1,M}]$ via polynomial approximation.** Recall that $\widehat{g}_M$ is the minimizer of the empirical error functional $\mathcal{E}_M(g)$ over the estimator space $\mathcal{G}_{r,K_M}^s$. Thus, we have

$$\mathcal{E}_M(\widehat{g}_M) - \mathcal{E}_M(g_\star) \leqslant \mathcal{E}_M(g_{\star,\mathcal{G}_r}) - \mathcal{E}_M(g_\star),$$

where $g_{\star,\mathcal{G}_{r,K_M}^s}$ is a minimizer in $\mathcal{G}_{r,K_M}^s$ attaining $\inf_{g\in\mathcal{G}_{r,K_M}^s}[\mathcal{E}_\infty(g) - \mathcal{E}_\infty(g_\star)]$ (see, (Györfi et al., 2006, Lemma 11.1)). Therefore,

$$\frac{1}{2}\mathbb{E}[T_{1,M}] = \mathbb{E}\left[\mathcal{E}_M(\widehat{g}_M) - \mathcal{E}_M(g_\star)\right]$$

$$\leqslant \mathbb{E}\left[\mathcal{E}_M(g_{\star,\mathcal{G}_{r,K_M}^s}) - \mathcal{E}_M(g_\star)\right]$$

$$= \mathcal{E}_\infty(g_{\star,\mathcal{G}_{r,K_M}^s}) - \mathcal{E}_\infty(g_\star) = \inf_{g\in\mathcal{G}_{r,K_M}^s}[\mathcal{E}_\infty(g) - \mathcal{E}_\infty(g_\star)]. \tag{B.6}$$

Note that

$$\mathcal{E}_\infty(g) - \mathcal{E}_\infty(g_\star) = \frac{1}{N}\mathbb{E}\left[\|R_{g_\star-g}[X] + \eta\|^2 - \|\eta\|^2\right]$$

$$= \frac{1}{N}\sum_{i=1}^N \mathbb{E}\left[\left|\frac{1}{N-1}\sum_{j=1,j\neq i}^N [g - g_\star](X_i, X_j)\right|^2\right]. \tag{B.7}$$

Applying Jensen's inequality to get

$$\frac{1}{N}\sum_{i=1}^N \mathbb{E}\left[\left|\frac{1}{N-1}\sum_{j=1,j\neq i}^N [g - g_\star](X_i, X_j)\right|^2\right] \leqslant \frac{1}{N(N-1)}\sum_{i=1}^N \sum_{j=1,j\neq i}^N \mathbb{E}\left[|[g - g_\star](X_i, X_j)|^2\right]$$

$$\tag{B.8}$$

and by the exchangeability assumption (A1), we have that the expectations are equal and thus

$$\frac{1}{2}\mathbb{E}[T_{1,M}] \leqslant \inf_{g \in \mathcal{G}^s_{r,K_M}} \|g - g_\star\|^2_{L^2_\rho}$$

$$= \inf_{\phi \in \Phi^s_{K_M}, \|A\|_{\mathrm{op}} \leqslant \bar{a}} \int |\phi(\langle x, y \rangle_A) - \phi_\star(\langle x, y \rangle_{A_\star})|^2 \mathrm{d}\rho(x,y) \,. \tag{B.9}$$

Next, setting $A = A_\star$ in equation B.9, it is clear that

$$\frac{1}{2}\mathbb{E}[T_{1,M}] \leqslant \inf_{\phi \in \Phi^s_{K_M}} \int |\phi(\langle x, y \rangle_{A_\star}) - \phi_\star(\langle x, y \rangle_{A_\star})|^2 \mathrm{d}\rho(x,y)$$

$$\leqslant \inf_{\phi \in \Phi^s_{K_M}} \left\{ \sup_{u \in [-\bar{a}, \bar{a}]} |\phi(u) - \phi_\star(u)|^2 \right\} \,.$$

Then, one can choose $\phi$ following the construction in (Györfi et al., 2006, Lemma 11.1) and that $\phi_\star \in C^\beta(L, \bar{a})$ which shows that there exists a piecewise polynomial function $f$ of degree $\beta$ or less with respect to an equidistant partition of $[-\bar{a}, \bar{a}]$ consisting of $K_M$ intervals of length $1/K_M$. For any $x, y \sim \rho$ and any matrix $A \in \mathbb{R}^{d \times d}$ such that $u = \langle x, y \rangle_A \in [-\bar{a}, \bar{a}]$, we will choose the dimension $K_M$ (to be specified later) so that

$$\sup_{u \in [-\bar{a}, \bar{a}]} |\phi(u) - \phi_\star(u)| \leqslant \frac{L\bar{a}^\beta}{\lfloor \beta \rfloor! K_M^\beta} \,.$$

We thus conclude that

$$\mathbb{E}[T_{1,M}] \leqslant \frac{2L^2 \bar{a}^{2\beta}}{(\lfloor \beta \rfloor!)^2 K_M^{2\beta}} \,. \tag{B.10}$$

**Step 3: Bounding $\mathbb{E}[T_{2,M}]$ via covering number estimates.** We introduce the following notations to simplify the presentation. Define $\Delta\mathcal{E}^{(i)}_M(g) := \mathcal{E}^{(i)}_M(g) - \mathcal{E}^{(i)}_M(g_\star)$. Also, we denote

$$\Delta\mathcal{E}^{(i)}_{\mathcal{D}_M}(\widehat{g}_M) := \mathbb{E}[|Y_i - R_{\widehat{g}_M}[X]_i|^2 \mid \mathcal{D}_M] - \mathbb{E}[|Y_i - R_{g_\star}[X]_i|^2] \,,$$

where $\widehat{g}_M \in \mathcal{G}^s_{r,K_M}$ depends on the samples $\mathcal{D}_M = \{(X^m, Y^m)\}^M_{m=1}$ and write similarly

$$\Delta\mathcal{E}^{(i)}_\infty(g) := \mathbb{E}[|Y_i - R_g[X]_i|^2] - \mathbb{E}[|Y_i - R_{g_\star}[X]_i|^2] \,,$$

for any $g \in \mathcal{G}^s_{r,K_M}$. Note that $\Delta\mathcal{E}^{(i)}_{\mathcal{D}_M}(g) = \Delta\mathcal{E}^{(i)}_\infty(g)$ for any (deterministic) $g \in \mathcal{G}^s_{r,K_M}$.

It is straightforward to observe that $\mathbb{E}[T_{2,M}]$ can be expressed as the average error per particle, that is, $\mathbb{E}[T_{2,M}] = \frac{1}{N}\sum^N_{i=1}\mathbb{E}[T^{(i)}_{2,M}]$ where $T^{(i)}_{2,M} := \Delta\mathcal{E}^{(i)}_{\mathcal{D}_M}(\widehat{g}_M) - T^{(i)}_{1,M}$ with $T^{(i)}_{1,M} = 2\Delta\mathcal{E}^{(i)}_M(\widehat{g}_M)$. To estimate $\mathbb{E}[T^{(i)}_{2,M}]$, it suffices to bound the following probability tail for the $i$-th particle

$$\mathbb{P}\left\{ T^{(i)}_{2,M} > t \right\} = \mathbb{P}\left\{ \Delta\mathcal{E}^{(i)}_{\mathcal{D}_M}(\widehat{g}_M) - \Delta\mathcal{E}^{(i)}_M(\widehat{g}_M) > \frac{1}{2}[t + \Delta\mathcal{E}^{(i)}_{\mathcal{D}_M}(\widehat{g}_M)] \right\}$$

$$\leqslant \mathbb{P}\left\{ \exists f \in \mathcal{G}^s_{r,K_M} : \Delta\mathcal{E}^{(i)}_{\mathcal{D}_M}(f) - \Delta\mathcal{E}^{(i)}_M(f) > \frac{1}{2}[t + \Delta\mathcal{E}^{(i)}_{\mathcal{D}_M}(f)] \right\}$$

$$= \mathbb{P}\left\{ \exists f \in \mathcal{G}^s_{r,K_M} : \Delta\mathcal{E}^{(i)}_\infty(f) - \Delta\mathcal{E}^{(i)}_M(f) > \frac{1}{2}[t + \Delta\mathcal{E}^{(i)}_\infty(f)] \right\} \,. \tag{B.11}$$

We first observe that the probability tail above depends on the joint distribution of all particles since the term $\Delta\mathcal{E}^{(i)}_M(f)$ in equation B.11 involves all particles. To bound the tail probability of $T^{(i)}_{2,M}$, we invoke Györfi et al. (2006, Theorem 11.4), which is applicable to classes of uniformly bounded functions. In our setting, this condition translates to the boundedness of the operator $R_g$. Specifically, if $\|g\|_\infty \leqslant B_\phi$, then for all $i \in [N]$, we have $|R_g[X]_i| \leqslant B_\phi$. Recall that $B_M :=$

$c_1 \log(M)$ and $\mathcal{C}_d := ([0,1]/\sqrt{d})^d$. Applying Theorem 11.4 in Györfi et al. (2006) to equation B.11 (with $\alpha = \beta = t/2$ and $\epsilon = 1/2$), we get for arbitrary $t \geqslant 1/M$

$$\mathbb{P}\left\{T_{2,M}^{(i)} > t\right\} \leqslant 14 \sup_{\{X^m \in \mathcal{C}_d^N\}_{m=1}^M} \mathcal{N}_1\left(\frac{t}{80B_M}, \mathcal{G}_{r,K_M}^s, \rho_M\right) e^{-\frac{tM}{24 \cdot 214 B_M^4}}$$

$$\leqslant 14 \sup_{\{X^m \in \mathcal{C}_d^N\}_{m=1}^M} \mathcal{N}_1\left(\frac{1}{80B_M M}, \mathcal{G}_{r,K_M}^s, \rho_M\right) e^{-\frac{tM}{24 \cdot 214 B_M^4}} \tag{B.12}$$

where $\mathcal{N}_1(\varepsilon, \mathcal{G}_{r,K_M}^s, \rho_M)$ is the empirical covering number with respect to the $L_\rho^1$ radius smaller than $\varepsilon$ over the function class $\mathcal{G}_{r,K_M}^s$.

Employing the identity $\mathbb{E}[X] = \int_0^\infty \mathbb{P}(X > t)\mathrm{d}t$ and the standard integral decomposition $\int_0^\infty = \int_0^\varepsilon + \int_\varepsilon^\infty$ with $\varepsilon$ to be determined, we get for $\varepsilon \geqslant 1/M$

$$\mathbb{E}[T_{2,M}^{(i)}] = \int_0^\infty \mathbb{P}(T_{2,M}^{(i)} > t)\mathrm{d}t \leqslant \varepsilon + \int_\varepsilon^\infty \mathbb{P}(T_{2,M}^{(i)} > t)\mathrm{d}t. \tag{B.13}$$

Then, substituting equation B.12 in equation B.13 leads to

$$\mathbb{E}[T_{2,M}^{(i)}] \leqslant \varepsilon + \int_\varepsilon^\infty 14 \sup_{\{X^m \in \mathcal{C}_d^N\}_{m=1}^M} \mathcal{N}_1\left(\frac{t}{80B_M}, \mathcal{G}_{r,K_M}^s, \rho_M\right) e^{-\frac{tM}{24 \cdot 214 B_M^4}} \mathrm{d}t. \tag{B.14}$$

Notice that we can bound the covering number by its value at $1/M$ since $\varepsilon \geqslant 1/M$ inside the integral in equation B.14 when $t \geqslant \varepsilon$. It then follows that

$$\mathbb{E}[T_{2,M}^{(i)}] \leqslant \varepsilon + 14 \sup_{\{X^m \in \mathcal{C}_d^N\}_{m=1}^M} \mathcal{N}_1\left(\frac{1}{80B_M M}, \mathcal{G}_{r,K_M}^s, \rho_M\right) \int_\varepsilon^\infty e^{-\frac{tM}{24 \cdot 214 \cdot B^4}} \mathrm{d}t. \tag{B.15}$$

We can now apply the estimate of covering number in equation B.22:

$$\mathbb{E}[T_{2,M}] = \frac{1}{N}\sum_{i=1}^N \mathbb{E}[T_{2,M}^{(i)}]$$

$$\leqslant \varepsilon + 42 \cdot (L_{1,M}M)^{2rd}(L_{2,M}M)^{2K_M(s+1)+2} \cdot \frac{L_{3,M}}{M} e^{-\frac{\varepsilon M}{L_{3,M}}} \tag{B.16}$$

with $L_{1,M} := 12r\bar{a}B_\phi \cdot 80B_M$, $L_{2,M} := 24eB_\phi \cdot 80B_M$ and $L_{3,M} := 24 \cdot 214 \cdot B_M^4$. Since the quantity $\varepsilon$ on the right-hand side of equation B.16 is arbitrary, we may tighten the bound by choosing

$$\varepsilon = \frac{L_{3,M}}{M} \cdot \log\left[42 \cdot (L_{1,M}M)^{2rd}(L_{2,M}M)^{2K_M(s+1)+2}\right],$$

which yields the desired upper bound:

$$\mathbb{E}[T_{2,M}] \leqslant \frac{L_{3,M}}{M}\Big[1 + \log(42) + 2rd\log(L_{1,M})$$

$$+ 2(K_M(s+1)+1) \cdot \log(L_{2,M}) + 2(K_M(s+1)+1+rd) \cdot \log(M)\Big]$$

$$\leqslant \frac{L_{3,M}(20K_M s + 5rd)\log(M)}{M} \tag{B.17}$$

when $M \geqslant \max(42 \cdot L_{1,M}^{2rd}, L_{2,M})$.

**Step 4: Bounding $\mathbb{E}[T_{3,M}]$ via sub-Gaussian property.** As $R_{g_\star}[X]_i \leqslant B_\phi < B_M$ and $R_{\hat{g}_M}[X]_i \leqslant B_\phi < B_M$ a.s. We assume that the noise $\eta$ is sub-Gaussian and that $R_g$ is bounded for any $g \in \mathcal{G}_r^\beta$. Thus, using Lemma 2 in Kohler & Mehnert (2011) with $Y_M$ and $B_M$ given above, one can obtain that with

$$\left|\mathbb{E}[T_{3,M}]\right| \leqslant \frac{1}{N}\sum_{i=1}^N \left|\mathbb{E}\big[|Y_{i,M} - R_{g_\star}[X_i]|^2\big] - \mathbb{E}\big[|Y_i - R_{g_\star}[X_i]|^2\big]\right|$$

$$+ \frac{1}{N}\sum_{i=1}^N \left|\mathbb{E}\big[|Y_{i,M} - R_{\hat{g}_M}[X_i]|^2\big] - \mathbb{E}\big[|Y_i - R_{\hat{g}_M}[X_i]|^2\big]\right|$$

$$\leqslant c_2 \frac{\log(M)}{M}, \tag{B.18}$$

for some constant $c_2 > 0$ independent of $M$ and $N$.

**Step 5: Deriving the upper optimal rate.** We now combine the bounds from equation B.10, equation B.16 and equation B.18, which control the terms $\mathbb{E}[T_{1,M}]$, $\mathbb{E}[T_{2,M}]$ and $\mathbb{E}[T_{3,M}]$, respectively, to obtain an upper bound on the total error in equation B.5:

$$
\mathbb{E}\Big[\|\widehat{g}_M - g_\star\|^2_{L^2_\rho}\Big]
$$

$$
\leqslant c_{\mathcal{H}}^{-1}\left(\frac{2L^2\bar{a}^{2\beta}}{(s!)^2 K_M^{2\beta}} + \frac{L_{3,M}(20K_M s + 5rd)\log(M)}{M} + c_2\frac{\log(M)}{M}\right) \tag{B.19}
$$

$$
\leqslant c_{\mathcal{H}}^{-1}\left(\frac{2L^2\bar{a}^{2\beta}}{(s!)^2 K_M^{2\beta}} + \frac{L_{3,M}20K_M s(1 + 5L_{3,M})\log(M)}{M} + c_2\frac{\log(M)}{M}\right)
$$

using the assumption of the theorem $rd \leqslant \left(\frac{M}{\log M}\right)^{\frac{1}{2\beta+1}}$ and setting the value of $K_M$ as

$$
K_M = \left\lfloor \frac{1}{20L_{3,M}s}\left(\frac{M}{\log M}\right)^{\frac{1}{2\beta+1}}\right\rfloor. \tag{B.20}
$$

A relatively straightforward choice of $K_M$ balances the terms in equation B.19 and leads to a desired upper bound. We note that a careful choice of $K_M$ may affect the constants and the power of $\log(M)$ in the upper bound.

Putting equation B.20 back into equation B.19 and noticing $c_{\mathcal{H}}^{-1} \leqslant N$, we get

$$
\mathbb{E}\left[\|\widehat{g}_M - g_\star\|^2_{L^2_\rho}\right] \leqslant c_{\mathcal{H}}^{-1}\Bigg[\frac{2L^2(20L_{3,M}s\bar{a})^{2\beta}}{(s!)^2}\left(\frac{\log M}{M}\right)^{\frac{2\beta}{2\beta+1}} + (1 + 5L_{3,M})\left(\frac{\log M}{M}\right)^{\frac{2\beta}{2\beta+1}}
$$

$$
+ \frac{c_2\log(M)}{M}\Bigg]
$$

$$
\leqslant N\left[\frac{2L^2(20sL_{3,M}\bar{a})^{2\beta}}{(s!)^2} + (1 + 5L_{3,M}) + c_2\right]\left(\frac{\log M}{M}\right)^{\frac{2\beta}{2\beta+1}} \tag{B.21}
$$

when $M \geqslant \max(42 \cdot L_{1,M}^{2rd}, L_{2,M})$ with $L_{1,M} := 12r\bar{a}B_\phi \cdot 80B_M$, $L_{2,M} := 24eB_\phi \cdot 80B_M$. Recalling that $L_{3,M} = 24 \cdot 214 \cdot B_M^4 = 24 \cdot 214 \cdot c_1 \cdot \log(M)$, we get from equation B.21 that

$$
\mathbb{E}\left[\|\widehat{g}_M - g_\star\|^2_{L^2_\rho}\right] \leqslant N\frac{2L^2(2024 \cdot 214 \cdot c_1 \cdot s\bar{a})^{2\beta}}{(s!)^2} \cdot \frac{[\log(M)]^{\frac{2\beta}{2\beta+1}+8\beta}}{M^{\frac{2\beta}{2\beta+1}}}
$$

$$
+ N[1 + 5 \cdot 24 \cdot 214 + c_2] \cdot \frac{(\log M)^{\frac{2\beta}{2\beta+1}+4}}{M^{\frac{2\beta}{2\beta+1}}}
$$

$$
\leqslant N\left[C_1^\beta\frac{L^2(s\bar{a})^{2\beta}}{(s!)^2} + C_2\right] \cdot \frac{[\log(M)]^{\frac{2\beta}{2\beta+1}+4\max(2\beta,1)}}{M^{\frac{2\beta}{2\beta+1}}}
$$

for some positive constant $C_1, C_2$. We complete the proof of Theorem 3.1 with $C_{N,L,\bar{a},\beta,s} = N[C_1^\beta\frac{L^2(s\bar{a})^{2\beta}}{(s!)^2} + C_2]$. □

Finally, to highlight the tradeoff between the parametric and the non-parametric part of the error, we present the following corollary. This corollary is directly derived from equation B.19 and equation B.20 without using the assumption $rd \leqslant \left(\frac{M}{\log M}\right)^{\frac{1}{2\beta+1}}$.

**Corollary B.2** *Consider the estimator $\widehat{g}_M$ defined in equation 3.1 computed on data $M$ i.i.d. observation satisfying Assumptions 2.1 and (B1). Then, for $\widehat{g}_M$ defined in equation 3.1 it holds that*

$$
\mathbb{E}\left[\|\widehat{g}_M - g_\star\|^2_{L^2_\rho}\right] \leqslant N\left[C_1^\beta\frac{L^2(s\bar{a})^{2\beta}}{(s!)^2} + C_2\right] \cdot \frac{[\log(M)]^{\frac{2\beta}{2\beta+1}+4\max(2\beta,1)}}{M^{\frac{2\beta}{2\beta+1}}} + C_3 rd \cdot \frac{(\log M)^2}{M}
$$

*where $C_1$, $C_2$ and $C_3$ are positive constants (maybe take different values than in the Theorem 3.1).*

### B.2 Auxiliary lemmas for the upper bound

Recall that the covering number $\mathcal{N}(\varepsilon, \mathcal{G}, d)$ is defined as the cardinality of the smallest $\varepsilon$-cover of $\mathcal{G}$ with respect to the metric $d$. When $d$ is the Euclidean metric, we omit it from the notation and simply write $\mathcal{N}(\varepsilon, \mathcal{G})$. It is also common to take $d$ to be an $L^p$-norm, either with respect to a probability measure $\rho$ or its empirical counterpart $\rho_M$. In these cases, we write

$$\mathcal{N}_p(\varepsilon, \mathcal{G}, \rho) := \mathcal{N}(\varepsilon, \mathcal{G}, \|\cdot\|_{L_\rho^p}), \quad \mathcal{N}_p(\varepsilon, \mathcal{G}, \rho_M) := \mathcal{N}(\varepsilon, \mathcal{G}, \|\cdot\|_{L_{\rho_M}^p}).$$

We next derive an upper bound for $\mathcal{N}_1(\varepsilon, \mathcal{G}^s_{r,K_M}, \rho_M)$, i.e., $p = 1$, by covering the matrix component and the functional component separately. Our argument combines the covering number estimates for matrices from Vershynin (2018) with the results of Györfi et al. (2006) for function classes.

**Lemma B.3** *Let $\mathcal{G}^s_{r,K_M}$ be defined in Definition 2.5. Assume that the sampled data $\{X_i^m\}_{i,m=1}^{N,M}$ are distributed according to Assumption 2.1. Then we have*

$$\mathcal{N}_1(\varepsilon, \mathcal{G}^s_{r,K_M}, \rho_M) \leqslant 3 \cdot \left(\frac{12r\bar{a}B_\phi}{\varepsilon}\right)^{2rd} \left(\frac{24eB_\phi}{\varepsilon}\right)^{2K_M(s+1)+2}. \tag{B.22}$$

**Proof.** Recall the matrix class defined in Definition 2.3:

$$\mathcal{A}_d(r, \bar{a}) := \{A \in \mathbb{R}^{d\times d} : \mathrm{rank}(A) \leqslant r, \|A\|_{\mathrm{op}} \leqslant \bar{a}\}.$$

Write $A = QK^\top$ via the truncated SVD, where $Q = U_r \Sigma_r^{1/2} \in \mathbb{R}^{d\times r}$ and $K = V_r \Sigma_r^{1/2} \in \mathbb{R}^{d\times r}$, with singular values belonging to $[0, \bar{a}]$, and $U_r, V_r^\top$ are semi unitary matrices of size $d \times r$, and $\Sigma_r$ is a diagonal matrix of size $r \times r$. Then

$$\|Q\|_F^2 = \mathrm{Tr}(\Sigma_r) \leqslant r\bar{a}, \qquad \|K\|_F^2 \leqslant r\bar{a}.$$

Indeed, let $\delta > 0$. The $\delta$-covering of the matrix class $\mathcal{Q}_{rd}(\sqrt{r\bar{a}}) := \{Q \in \mathbb{R}^{d\times r} : \|Q\|_F \leqslant \sqrt{r\bar{a}}\}$ is equivalent to the $\delta$-covering of $B_{rd}(\sqrt{r\bar{a}})$, a centered ball with radius $\sqrt{r\bar{a}}$ in $\mathbb{R}^{rd}$ and (Vershynin, 2018, Corollary 4.2.11) implies that

$$n = \mathcal{N}(\epsilon, \mathcal{Q}_{rd}(\sqrt{r\bar{a}}), \|\cdot\|_F) = \mathcal{N}(\epsilon, B_{rd}(\sqrt{r\bar{a}})) \leqslant \left(\frac{3\sqrt{r\bar{a}}}{\epsilon}\right)^{rd}. \tag{B.23}$$

Notice that for $A_1 = Q_1 K_1^\top$ and $A_2 = Q_2 K_2^\top$ with $\{Q_i \in \mathcal{Q}_{rd}(\sqrt{r\bar{a}}), K_i \in \mathcal{Q}_{rd}(\sqrt{r\bar{a}})\}_{i=1}^2$ such that $\|Q_1 - Q_2\|_F \leqslant \delta/(2\sqrt{\bar{a}})$, $\|K_1 - K_2\|_F \leqslant \delta/(2\sqrt{\bar{a}})$, we have

$$\|A_1 - A_2\|_{\mathrm{op}} = \|Q_1 K_1^\top - Q_2 K_2^\top\|_{\mathrm{op}} \leqslant \|Q_1\|_{\mathrm{op}} \|K_1 - K_2\|_F + \|Q_1 - Q_2\|_F \|K_2\|_{\mathrm{op}} \leqslant \delta.$$

Moreover, by Assumption 2.1 that $X_i, X_j$ lie within the unit ball and the assumption that $\phi \in \Phi^s_{K_M}$, a degree-$s$ piecewise-polynomial approximation with $K_M$ intervals, we get:

$$\left|\phi(\langle x, y\rangle_{A_1}) - \phi(\langle x, y\rangle_{A_2})\right| \leqslant B_\phi \|A_1 - A_2\|_{\mathrm{op}} \leqslant B_\phi \delta$$

since $\|x\|, \|y\| \leqslant 1$. This proves that if $A_1, A_2$ are within $\delta$ in operator norm, the corresponding functions differ by at most $B_\phi \delta$. Thus,

$$\mathcal{N}_1(2B_\phi \delta, \mathcal{G}^s_{r,K_M}, \rho_M) \leqslant \sum_{i,j\neq i}^n \mathcal{N}_1(B_\phi \delta, \{\phi(\langle x, y\rangle_{Q_i K_j^\top}) : \phi \in \Phi^s_{K_M}\}, \rho_M). \tag{B.24}$$

On the other hand, (Györfi et al., 2006, Theorem. 9.4–9.5) shows the following bound

$$\mathcal{N}_1(B_\phi \delta, \Phi^s_{K_M}, \rho_M) \leqslant 3\left(\frac{6e(B_\phi + 1)}{B_\phi \delta}\right)^{2K_M(s+1)+2} \leqslant 3\left(\frac{12e}{\delta}\right)^{2K_M(s+1)+2}$$

for the empirical measure $\rho_M$ in Definition 2.1 with $\{X^m \in \mathcal{C}_d\}_{m=1}^M$. Putting it back to equation B.24, we obtain that

$$\mathcal{N}_1(2B_\phi \delta, \mathcal{G}^s_{r,K_M}, \rho_M) \leqslant 3 \cdot \left(\frac{6r\bar{a}}{\delta}\right)^{2rd} \left(\frac{12e}{\delta}\right)^{2K_M(s+1)+2}. \tag{B.25}$$

Now re-parameterize by $\varepsilon = 2B_\phi \delta$, i.e. $\delta = \varepsilon/2B_\phi$, and absorb constants in equation B.25. This gives our desired estimate in equation B.22. $\qquad\square$

**Remark B.4** *As a by-product, we show that a $\frac{\delta}{2\sqrt{r\bar{a}}}$-cover for $Q$ and $K$ induces a $\delta$-cover for $\mathcal{A}_d(r, \bar{a})$ in operator norm. Taking all pairs $Q_i K_j^\top$ and substituting $\epsilon = \frac{\delta}{2\sqrt{r\bar{a}}}$ in equation B.23 give that*

$$
\mathcal{N}(\delta, \mathcal{A}(r, \bar{a}), \|\cdot\|_{\mathrm{op}}) \leqslant \left(\frac{6r\bar{a}}{\delta}\right)^{2rd}.
$$

## C    Appendix: Lower bound proofs

**Lemma C.1 (Continuous density of the bilinear form.)** *Let $(X, Y) \in \mathbb{R}^{2d}$ have a joint density $p \in L^1(D)$ with $D \subset \mathbb{R}^{2d}$ being a bounded open set. Let $A \in \mathbb{R}^{d \times d}$ have rank $r \geqslant 1$, and define $U = X^\top A Y$. Then:*

(i) *(Existence) For every $r \geqslant 1$, the law of $U$ is absolutely continuous with respect to Lebesgue measure on $\mathbb{R}$ with a density denoted by $p_U$.*

(ii) *(Continuity) If $r \geqslant 2$, then $p_U \in C(\mathbb{R})$.*

Note that $r \geqslant 2$ is *sharp* for $p_U$ to be continuous: for $r = 1$, continuity at $0$ may not hold: if $X, Y$ are independent standard Gaussian in $\mathbb{R}$ and $A = I$, then $U = XY$ has density $p_U(u) = \frac{1}{\pi} K_0(|u|)$, where $K_0(x) \sim -\log x$ as $x \downarrow 0$, so $p_U$ is singular at $0$.

**Proof.** The proof consists of three steps: reduction to a canonical quadratic form on $\mathbb{R}^{2r}$, existence of the density, and continuity.

**Step 1. Reduction to the canonical quadratic form on $\mathbb{R}^{2r}$.** Let $A = W\Sigma V^\top$ be a singular value decomposition with $\Sigma = \mathrm{diag}(\sigma_1, \ldots, \sigma_r, 0, \ldots, 0)$, where $\sigma_i > 0$ and $W, V \in \mathbb{R}^{d \times d}$ are orthonormal. Set $\alpha := W^\top X$, $\beta := V^\top Y$. Orthonormal changes preserve absolute continuity, so $(\alpha, \beta)$ has a joint density $\widetilde{p}(\alpha, \beta) = p(W\alpha, V\beta)$ with support $\widetilde{D} = \{(\alpha, \beta) = (W^{-1}x, V^{-1}y) : (x, y) \in D\}$, which is a bounded subset in $\mathbb{R}^{2d}$. Split $\alpha = (\alpha^{(r)}, \alpha^\perp)$ and $\beta = (\beta^{(r)}, \beta^\perp)$, where the superscript $(r)$ denotes the first $r$ coordinates. Then

$$
U = X^\top A Y = \sum_{i=1}^{r} \sigma_i \, \alpha_i^{(r)} \, \beta_i^{(r)}.
$$

Integrating out $(\alpha^\perp, \beta^\perp)$ yields a marginal density $q \in L^1(\mathbb{R}^{2r})$ for $Z := (\alpha^{(r)}, \beta^{(r)})$ and it has a bounded support, which we denote by $\Omega$. Thus it suffices to work in $\mathbb{R}^{2r}$ with

$$
\Phi(\alpha, \beta) := \sum_{i=1}^{r} \sigma_i \alpha_i \beta_i, \qquad U = \Phi(Z), \qquad Z \sim q \in L^1(\Omega).
$$

**Step 2. Existence by the coarea formula.** For any bounded measurable test function $\varphi : \mathbb{R} \to \mathbb{R}$,

$$
\mathbb{E}[\varphi(U)] = \int_\Omega \varphi(\Phi(z)) \, q(z) \, dz. \tag{C.1}
$$

Note that $\Phi$ has gradient $\nabla\Phi(\alpha, \beta) = (\sigma_1\beta_1, \ldots, \sigma_r\beta_r, \sigma_1\alpha_1, \ldots, \sigma_r\alpha_r)$, which is Lipschitz continuous on $\Omega$. Applying the Coarea formula (i.e., for any $f : \Omega \subset \mathbb{R}^n \to \mathbb{R}$ Lipschitz and $g \in L^1_{\mathrm{loc}}(\mathbb{R}^n)$, $\int_{\mathbb{R}^n} g(z) \, |\nabla f(z)| \, dz = \int_{\mathbb{R}} \left( \int_{f^{-1}(u)} g(z) \, d\mathcal{H}^{n-1}(z) \right) du$, where $\mathcal{H}^{n-1}(z)$ denotes the Hausdorff measure, see, e.g., Evans (2018)) to $f = \Phi$ with $g(z) = q(z) \, \varphi(\Phi(z))/|\nabla\Phi(z)|$ gives

$$
\int_\Omega \varphi(\Phi(z)) \, q(z) \, dz = \int_{\mathbb{R}} \left( \int_{\Phi^{-1}(u)} \frac{q(z)}{|\nabla\Phi(z)|} \, d\mathcal{H}^{2r-1}(z) \right) \varphi(u) \, du.
$$

Hence, $p_U(u) = \int_{\Phi^{-1}(u)} \frac{q(z)}{|\nabla\Phi(z)|} \, d\mathcal{H}^{2r-1}(z)$ for $u \neq 0$.

Note that under the change of variables $z = \sqrt{|u|} \, w$, the Hausdorff surface measure scales by $|u|^{(2r-1)/2}$ and $|\nabla\Phi|$ by $|u|^{1/2}$. Then, for $u \neq 0$, the above equation can be written as

$$
\int_\Omega \varphi(\Phi(z)) \, q(z) \, dz = \int_{\mathbb{R}} |u|^{r-1} \left( \int_{\Phi(w)=\mathrm{sign}(u)} \frac{q(\sqrt{|u|}\, w)}{|\nabla\Phi(w)|} \, d\mathcal{H}^{2r-1}(w) \right) \varphi(u) \, du. \tag{C.2}
$$

Comparing equation C.1 and equation C.2, the push-forward measure is absolutely continuous with density

$$p_U(u) = |u|^{r-1} \int_{\Phi(w)=\text{sign}(u)} \frac{q(\sqrt{|u|}\,w)}{|\nabla\Phi(w)|}\, d\mathcal{H}^{2r-1}(w) \tag{C.3}$$

for all $u \neq 0$. This proves (i) for all $r \geqslant 1$.

**Step 3. Continuity.** Let $\xi_U(t) = \mathbb{E}[e^{itU}] = \int_{\mathbb{R}^{2r}} e^{it\Phi(z)} q(z)\, dz$ be the characteristic function. The phase $t\Phi(z)$ is a non-degenerate quadratic form with constant Hessian $H = t \begin{pmatrix} 0 & \Sigma \\ \Sigma & 0 \end{pmatrix}$ (of full rank $2r$ ). By the standard stationary phase bound for quadratic phases (see, e.g., (Sogge, 2017, Theorem 1.1.4))

$$|\xi_U(t)| \leqslant C\,(1+|t|)^{-r},$$

with $C$ depending on $q$ (e.g. if $q \in C_c^\infty$ , then $C$ depends on a finite number of derivatives; and it extends to general $q \in L^1$ since $C_c^\infty$ is dense in $L^1$). Hence, if $r \geqslant 2$ , then $\xi_U \in L^1(\mathbb{R})$ and Fourier inversion yields a bounded continuous density $p_U(u) = \frac{1}{2\pi} \int_{\mathbb{R}} e^{-itu}\xi_U(t)\, dt$. □

Next, we provide the proof of Lemma 4.1.

**Proof of Lemma 4.1.** Consider

$$U_{ij} = X_i^\top A_\star X_j, \qquad V_{ij} = X_i^\top \hat{A} X_j,$$

so that $\hat{g}(X_i, X_j) = \hat{\phi}(V_{ij})$ and $g^\star(X_i, X_j) = \phi^\star(U_{ij})$. Recall that $p_{U_{ij}}$ is the density of $U_{ij}$ and $p_U = \frac{1}{N(N-1)} \sum_{i,j:i\neq j} p_{U_{ij}}$. Also, recall that the following functions are defined in equation 4.2:

$$\hat{\psi}_{ij}(u) := \mathbb{E}[\hat{\phi}(V_{ij})|U_{ij} = u], \quad \hat{\psi}(u) := \sum_{i=1}^N \sum_{j=1,j\neq i}^N \frac{p_{U_{ij}}(u)}{N(N-1)p_U(u)}\hat{\psi}_{ij}(u).$$

Since $\sum_{i=1}^N \sum_{j=1,j\neq i}^N \frac{p_{U_{ij}}(u)}{N(N-1)p_U(u)} = 1$, we have, by applying Jensen's inequality,

$$\left|\hat{\psi}(u) - \phi_\star(u)\right|^2 = \left|\sum_{i=1}^N \sum_{j=1,j\neq i}^N \frac{p_{U_{ij}}(u)}{N(N-1)p_U(u)}\hat{\psi}_{ij}(u) - \phi_\star(u)\right|^2$$

$$\leqslant \sum_{i=1}^N \sum_{j=1,j\neq i}^N \frac{p_{U_{ij}}(u)}{N(N-1)p_U(u)}\left|\hat{\psi}_{ij}(u) - \phi_\star(u)\right|^2. \tag{C.4}$$

Also, by applying Jensen's inequality to the conditional expectation, we have

$$\mathbb{E}\left[\left|\hat{\phi}(V_{ij}) - \phi_\star(U_{ij})\right|^2\right] = \mathbb{E}\left[\mathbb{E}\left[\left|\hat{\phi}(V_{ij}) - \phi_\star(U_{ij})\right|^2 |U_{ij}\right]\right]$$

$$\geqslant \mathbb{E}\left[\left|\mathbb{E}[\hat{\phi}(V_{ij}) - \phi_\star(U_{ij})|U_{ij}]\right|^2\right] = \int_{-\bar{a}}^{\bar{a}}\left|\hat{\psi}_{ij}(u) - \phi_\star(u)\right|^2 p_{U_{ij}}(u)du.$$

Averaging over the pairs as in equation C.4, we have

$$\frac{1}{N(N-1)} \sum_{i=1}^N \sum_{j=1,j\neq i}^N \mathbb{E}\left[\left|\hat{\phi}(V_{ij}) - \phi_\star(U_{ij})\right|^2\right]$$

$$\geqslant \int_{-\bar{a}}^{\bar{a}} \frac{1}{N(N-1)} \sum_{i=1}^N \sum_{j=1,j\neq i}^N \left|\hat{\psi}_{ij}(u) - \phi_\star(u)\right|^2 \frac{p_{U_{ij}}(u)}{p_U(u)}p_U(u)du$$

$$\geqslant \int_{-\bar{a}}^{\bar{a}}\left|\hat{\psi}(u) - \phi_\star(u)\right|^2 p_U(u)\, du,$$

which is the desired inequality. □

**Proof of Lemma 4.2** We construct $\bar{K} = \lceil c_{0,N} M^{\frac{1}{2\beta+1}}\rceil$ disjoint equidistant intervals.

$$\{\Delta_\ell = (r_\ell - h_M, r_\ell + h_M)\}_{\ell=1}^{\bar{K}}, \quad \text{with } h_M = \frac{L_0}{8n_0\bar{K}}, \tag{C.5}$$

where $\{r_\ell\}$, $n_0$ and $L_0$ are specific values that will be determined below. We will define the intervals by separating into two cases: one where the density of $p_U$ is bounded below by $\underline{a}_0 > 0$ and one where it is not.

If $p_U(u) \geq \underline{a}_0 > 0$, we can simply use the uniform partition of $\mathrm{supp}(p_U)$ to obtain the desired $\{\Delta_\ell\}$. That is, we set $n_0 = 1$, $L_0 = 4$, and $r_\ell = -\bar{a} + (2\ell - 1)h_M$. If $p_U$ is not bounded away from zero, we shall build the partition based on its continuity. Since $p_U$ is continuous on $[-\bar{a}, \bar{a}]$, the constant $a_0 = \sup_{x \in [-\bar{a}, \bar{a}]} p_U(x)$ exists, now consider $\underline{a}_0 < a_0 \wedge 1$. We can construct the $\bar{K}$ intervals described in equation C.5 which satisfy the following $\bigcup_\ell \Delta_\ell \subset A_0 := \{u \in [-\bar{a}, \bar{a}] : p_U(u) > \underline{a}_0\}$.

Let $L_0 := \frac{1 - 2\underline{a}_0}{a_0 - \underline{a}_0}$. Since for all $u \in A_0$, $p_U(u) \leq a_0$ and for all $u \in A_0^c$, $p_U(u) \leq \underline{a}_0$, together with the fact that $1 = \int_{A_0} p_U(u)du + \int_{A_0^c} p_U(u)du$, we get:

$$1 \leq a_0 \mathrm{Leb}(A_0) + \underline{a}_0(2\bar{a} - \mathrm{Leb}(A_0)) \Rightarrow L_0 \leq \mathrm{Leb}(A_0) \leq 2\bar{a}. \tag{C.6}$$

Also, note that the set $A_0$ is open by continuity of $p_U$. Thus, there exist disjoint intervals $(a_j, b_j)$ such that $A_0 = \bigcup_{j=1}^\infty (a_j, b_j)$. Without loss of generality, we assume that these intervals are descendingly ordered according to their length $b_j - a_j$. Let

$$n_0 = \min\{n : \sum_{j=1}^n (b_j - a_j) > \frac{L_0}{2}\}. \tag{C.7}$$

One can see that $n_0 > 1$. Now, we construct the first $n_1$ disjoint intervals $\{\Delta_\ell = (r_\ell - h_M, r_\ell + h_M)\}_{\ell=1}^{n_1} \subset (a_1, b_1)$ such that $r_\ell = a_1 + \ell h_M$ and $n_1 = \lfloor \frac{b_1 - a_1}{2h_M} \rfloor$. If $n_1 \geq \bar{K}$, we stop. Otherwise, we construct additional disjoint intervals $\{\Delta_\ell = (r_\ell - h_M, r_\ell + h_M)\}_{\ell=n_1+1}^{n_1+n_2} \subset (a_2, b_2)$ similarly, and continue to $(a_j, b_j)$ until obtaining $\bar{K}$ intervals $\{\Delta_\ell\}$.

To show that we will at least obtain $\bar{K}$ such intervals, we show that $K_\star \geq \bar{K}$, where $K_\star$ is the total number of intervals $\{\Delta_\ell\}_{\ell=1}^{K_\star}$. Since the Lebesgue measure of $(a_j, b_j) \backslash \bigcup_{\ell=1}^{K_\star} \Delta_\ell$ is less than $2h_M$ for each $j$, the Lebesgue measure of the uncovered parts $\bigcup_{j=1}^{n_0}(a_j, b_j) \backslash (\bigcup_{\ell=1}^{K_\star} \Delta_\ell)$ is at most $2n_0 h_M$.

Thus, by equation C.7 the intervals $\{\Delta_\ell\}_{\ell=1}^{K_\star}$ must have a total length no less than $\frac{L_0}{2} - 2n_0 h_M$. And since each of them is in length of $2h_M$ the total number must satisfy:

$$K_\star \geq (\frac{L_0}{2} - 2n_0 h_M)/(2h_M)$$

and plugging in the definition of $h_M$ from equation C.5 we get:

$$K_\star \geq 2\bar{K}n_0 - n_0 \geq \bar{K}.$$

Now we construct hypothesis functions satisfying Conditions (D1)–(D3). We first define $2^{\bar{K}}$ functions, from which we will select a subset of $2s$-separated hypothesis functions,

$$\phi_\omega(u) = \sum_{\ell=1}^{\bar{K}} \omega_\ell \psi_{\ell,M}(u), \quad \omega = (\omega_1, \cdots, \omega_{\bar{K}}) \in \{0, 1\}^{\bar{K}},$$

where the basis functions are

$$\psi_{\ell,M}(u) := Lh_M^\beta \psi\left(\frac{u - r_\ell}{h_M}\right), \quad u \in [-\bar{a}, \bar{a}] \tag{C.8}$$

with $\psi(u) = e^{-\frac{1}{1-(2u)^2}} \mathbf{1}_{|u| \leq 1/2}$. Note that the support of $\psi_{\ell,M}(u)$ is $\Delta_\ell$, and $\int_{\Delta_\ell} |\psi_{\ell,M}(u)|^2 du = L^2 h_M^{2\beta+1} \|\psi\|_2^2$. By definition, these hypothesis functions satisfy Condition (D1), i.e., they are Holder continuous and

$$\|\psi_{\ell,M}\|_\infty \leq Lh_M^\beta \leq L\|p_U\|_\infty^{-\beta} \leq L(2\bar{a})^{-\beta} \leq B_\phi$$

since $h_M = \frac{L_0}{8n_0 \bar{K}} < L_0 \leq \frac{1}{a_0}$ with $a_0 = \|p_U\|_\infty$ and $\|p_U\|_\infty \leq \frac{1}{2\bar{a}}$.

Then, denoting $\phi_k(x) = \phi_{\omega^{(k)}}(x)$, we proceed to verify Conditions (D2)–(D3). Next, we select a subset of $2s_{N,M}$-separated functions $\{\phi_{k,M} := \phi_{\omega^{(k)}}\}_{k=1}^K$ satisfying Condition (D2), i.e., $\|\phi_{\omega^{(k)}} -$

$\phi_{\omega^{(k')}}\|_{L^2_{p_U}} \geq 2s_{N,M}$ for any $k \neq k' \in \{1, \ldots, K\}$. Here $s_{N,M} = C_1 c_{0,N}^{-\beta} M^{-\frac{\beta}{2\beta+1}}$ with $C_1$ being a positive constant to be determined below. Since $\Delta_\ell = \mathrm{supp}(\psi_{\ell,M}) \subseteq \Delta_\ell$ are disjoint, we have

$$\|\phi_\omega - \phi_{\omega'}\|_{L^2_{p_U}} = \left( \int_{\mathbb{R}} \left| \sum_{\ell=1}^{\bar{K}} (\omega_\ell - \omega'_\ell)\psi_{\ell,M}(u) \right|^2 p_U(u)du \right)^{\frac{1}{2}}$$

$$= \left( \sum_{\ell=1}^{\bar{K}} (\omega_\ell - \omega'_\ell)^2 \int_{\Delta_\ell} |\psi_{\ell,M}(u)|^2 p_U(u)du \right)^{\frac{1}{2}}.$$

Since $p_U(u) \geq \underline{a}_0$ over each $\Delta_\ell$, we have

$$\int_{\Delta_\ell} |\psi_{\ell,M}(u)|^2 p_U(u)du \geq \underline{a}_0 \int_{\Delta_\ell} |\psi_{\ell,M}(u)|^2 du = \underline{a}_0 L^2 h_M^{2\beta+1} \|\psi\|_2^2.$$

Applying the Varshamov-Gilbert bound (Tsybakov, 2008, Lemma 2.9), one can obtain a subset $\{\omega^{(k)}\}_{k=1}^K$ with $K \geq 2^{\bar{K}/8}$ such that $\sum_\ell^{\bar{K}} (\omega_\ell^{(k)} - \omega_\ell^{(k')})^2 \geq \frac{\bar{K}}{8}$ for any $k \neq k' \in \{1, \ldots, K\}$. Thus,

$$\|\phi_\omega - \phi_{\omega'}\|_{L^2_{p_U}} \geq \sqrt{\underline{a}_0} L \|\psi\|_2 \sqrt{\frac{\bar{K}}{8}} \left( \frac{L_0}{8n_0 \bar{K}} \right)^{\beta+1/2}$$

$$= \sqrt{\underline{a}_0} L \|\psi\|_2 \frac{\bar{K}^{1/2}}{2\sqrt{2}} \left( \frac{L_0}{8n_0} \right)^{\beta+1/2} \bar{K}^{-(\beta+1/2)}$$

$$= \left( \frac{\sqrt{\underline{a}_0} L \|\psi\|_2}{2\sqrt{2}} \left( \frac{L_0}{8n_0} \right)^{\beta+1/2} \right) \bar{K}^{-\beta} = s_{N,M}$$

where $s_{N,M} = C_1 c_{0,N}^{-\beta} M^{-\frac{\beta}{2\beta+1}}$ with

$$C_1 := \frac{\sqrt{\underline{a}_0} L \|\psi\|_2}{4\sqrt{2}} \left( \frac{L_0}{8n_0} \right)^{\beta+1/2}.$$

To verify condition (D3) for each fixed dataset $\{X^m\}_{m=1}^M$, we first compute the Kullback-Leibler (KL) divergence. Define $u_{ij}^m := (X_i^m)^\top A X_j^m$. Then for each $m$,

$$R_\phi[X^m]_i = \frac{1}{N-1} \sum_{j \neq i} \phi(u_{ij}^m).$$

Under the hypothesis $\phi_{k,M}$, the density of the outputs $\{Y^m\}_{m=1}^M$ is

$$p_k(y^1, \ldots, y^M) = \prod_{m=1}^M p_\eta \left( y^m - R_{\phi_{k,M}}[X^m] \right),$$

where $y^m \in \mathbb{R}^d$ represents the observed output $Y^m$. By definition of KL divergence and the i.i.d. noise assumption,

$$D_{\mathrm{KL}}(\bar{\mathbb{P}}_k, \bar{\mathbb{P}}_0) = \int \cdots \int \log \prod_{m=1}^M \frac{p_\eta(y^m)}{p_\eta\left( y^m + R_{\phi_{k,M}}[X^m] \right)} \prod_{m=1}^M p_\eta(y^m) dy^m.$$

This simplifies to

$$D_{\mathrm{KL}}(\bar{\mathbb{P}}_k, \bar{\mathbb{P}}_0) = \sum_{m=1}^M \int_{\mathbb{R}^d} \log \left[ \frac{p_\eta(y^m)}{p_\eta\left( y^m + R_{\phi_{k,M}}[X^m] \right)} \right] p_\eta(y^m) dy^m.$$

Finally, by the noise smoothness assumption 2.2, for each $m$,

$$\int p_\eta(y) \log \left[ \frac{p_\eta(y)}{p_\eta(y+v)} \right] dy \leq c_\eta \|v\|^2,$$

where $v = R_{\phi_{k,M}}[X^m]$. Summing over $m = 1, \ldots, M$ yields

$$D_{\mathrm{KL}}(\mathbb{P}_k, \mathbb{P}_0) \leqslant c_\eta \sum_{m=1}^{M} \| R_{\phi_{k,M}}[X^m] \|^2, \tag{C.9}$$

Employing Jensen's inequality, we have

$$\| R_{\phi_{k,M}}[X^m] \|^2 = \sum_{i=1}^{N} \Big( \frac{1}{N-1} \sum_{j \neq i} \phi_{k,M}(u_{ij}^m) \Big)^2 \leqslant \sum_{i=1}^{N} \frac{1}{N-1} \sum_{j \neq i} |\phi_{k,M}(u_{ij}^m)|^2 = \frac{1}{N-1} \sum_{i=1}^{N} \sum_{j \neq i} |\phi_{k,M}(u_{ij}^m)|^2. \tag{C.10}$$

Recalling that $\phi_{k,M}(u_{ij}^m) = \sum_{\ell=1}^{\bar{K}} \omega_\ell^{(k)} \psi_{\ell,M}(u_{ij}^m)$, where $\mathrm{supp}(\psi_{\ell,M}) \subseteq \Delta_\ell$ are disjoint and $|\psi_{\ell,M}(u_{ij}^m)| = L h_M^\beta \psi\Big( \frac{u_{ij}^m - r_\ell}{h_M} \Big) \leqslant L h_M^\beta \|\psi\|_\infty \mathbf{1}_{\{u_{ij}^m \in \Delta_\ell\}}$, we have

$$|\phi_{k,M}(u_{ij}^m)|^2 = \sum_{\ell=1}^{\bar{K}} \omega_\ell^{(k)} |\psi_{\ell,M}(u_{ij}^m)|^2 \leqslant L^2 h_M^{2\beta} \|\psi\|_\infty^2 \sum_{\ell=1}^{\bar{K}} \mathbf{1}_{\{u_{ij}^m \in \Delta_\ell\}}, \tag{C.11}$$

where we have used the fact that $0 \leqslant \omega_\ell^{(k)} \leqslant 1$. By plugging in both equation C.10 and equation C.11 into equation C.9, we obtain

$$D_{\mathrm{KL}}(\bar{\mathbb{P}}_k, \bar{\mathbb{P}}_0) \leqslant \frac{c_\eta}{N-1} \sum_{m=1}^{M} \sum_{i=1}^{N} \sum_{j \neq i} \Big( L^2 h_M^{2\beta} \|\psi\|_\infty^2 \sum_{\ell=1}^{\bar{K}} \mathbf{1}_{\{u_{ij}^m \in \Delta_\ell\}} \Big)$$

$$\leqslant \frac{c_\eta L^2 \|\psi\|_\infty^2 h_M^{2\beta}}{N-1} \sum_{i,j,m} \Big( \sum_{\ell=1}^{\bar{K}} \mathbf{1}_{\{u_{ij}^m \in \Delta_\ell\}} \Big).$$

Since the intervals $\{\Delta_\ell\}$ are disjoint, the inner sum is at most 1. The total sum over $i, j, m$ is therefore bounded by $N^2 M$, which gives:

$$D_{\mathrm{KL}}(\bar{\mathbb{P}}_k, \bar{\mathbb{P}}_0) \leqslant c_\eta L^2 \|\psi\|_\infty^2 N M h_M^{2\beta}.$$

Hence, by assigning $h_M = \frac{L_0}{8 n_0 \bar{K}}$ from equation C.5 and $\bar{K} = \lceil c_{0,N} M^{\frac{1}{2\beta+1}} \rceil$, we obtain

$$\frac{1}{K} \sum_{k=1}^{K} D_{\mathrm{KL}}(\bar{\mathbb{P}}_k, \bar{\mathbb{P}}_0) \leqslant \Big( c_\eta L^2 \|\psi\|_\infty^2 \Big( \frac{L_0}{8 n_0} \Big)^{2\beta} \Big) N \Big( \frac{\bar{K}}{c_{0,N}} \Big)^{2\beta+1} \bar{K}^{-2\beta}$$

$$= \Big( \frac{c_\eta L^2 \|\psi\|_\infty^2 N}{c_{0,N}^{2\beta+1}} \Big( \frac{L_0}{8 n_0} \Big)^{2\beta} \Big) \bar{K} \leqslant \alpha \log K$$

with $\alpha = \Big( \frac{c_\eta L^2 \|\psi\|_\infty^2 N}{c_{0,N}^{2\beta+1}} \Big( \frac{L_0}{8 n_0} \Big)^{2\beta} \Big) \frac{8}{\log 2}$ since $K \geqslant 2^{\bar{K}/8}$. Thus, for condition (D3) to hold, i.e., $\alpha < 1/8$, we need

$$c_{0,N}^{2\beta+1} \geqslant 64 c_\eta L^2 \|\psi\|_\infty^2 N \Big( \frac{L_0}{8 n_0} \Big)^{2\beta}.$$

Following $c_{0,N} = C_0 N^{\frac{1}{2\beta+1}}$, it suffices to set $C_0$ to be

$$C_0 := (32 c_\eta L^2 \|\psi\|_\infty^2 \Big( \frac{L_0}{8 n_0} \Big)^{2\beta})^{\frac{1}{2\beta+1}}.$$

$\square$

To prove the lower bound minimax rate, we will use the following lower bound for hypothesis test error, see e.g., Proposition 2.3 Tsybakov (2008) or Lemma 4.3 in Wang et al. (2025).

**Lemma C.2 (Lower bound for hypothesis test error )** *Let $\Theta = \{\theta_k\}_{k=0}^K$ with $K \geqslant 2$ be a set of $2s$-separated hypotheses, i.e., $d(\theta_k, \theta_{k'}) \geqslant 2s > 0$ for all $0 \leqslant k < k' \leqslant K$, for a given metric $d$ on $\Theta$. Denote $\mathbb{P}_k = \mathbb{P}_{\theta_k}$ and suppose they satisfy $\mathbb{P}_k \ll \mathbb{P}_0$ for each $k \geqslant 1$ and*

$$\frac{1}{K+1} \sum_{k=1}^K D_{\mathrm{KL}}(\mathbb{P}_k, \mathbb{P}_0) \leqslant \alpha \log(K), \quad \text{with } 0 < \alpha < 1/8. \tag{C.12}$$

*Then, the average probability of the hypothesis testing error has a lower bound:*

$$\inf_{k_{\mathrm{test}}} \frac{1}{K+1} \sum_{k=0}^K \mathbb{P}_k\big(k_{\mathrm{test}} \neq k\big) \geqslant \frac{\log(K+1) - \log(2)}{\log(K)} - \alpha, \tag{C.13}$$

*where $\inf_{k_{\mathrm{test}}}$ denotes the infimum over all tests.*

**Proof of Theorem 4.3** We aim to apply Tsybakov's method to simplify probability bounds by considering a finite set of hypothesis functions. Reducing the supremum over $\mathcal{C}^\beta(L, \bar{a})$ to the finite set of hypothesis functions, and applying the Markov inequality, we obtain

$$\sup_{\substack{\phi_\star \in \mathcal{C}^\beta(L, \bar{a}) \\ \|\phi_\star\|_\infty \leqslant B_\phi}} \mathbb{E}_{\phi_\star}\left[\|\widehat{\phi}_M - \phi_\star\|_{L^2_{p_U}}^2\right]$$

$$\geqslant \max_{\phi_{k,M} \in \{\phi_{0,M},\dots,\phi_{K,M}\}} \mathbb{E}_{\phi_{k,M}}\left[\|\widehat{\phi}_M - \phi_{k,M}\|_{L^2_{p_U}}^2\right]$$

$$\geqslant \max_{\phi_{k,M} \in \{\phi_{0,M},\dots,\phi_{K,M}\}} s_{N,M}^2 \mathbb{P}_{\phi_{k,M}}\left[\|\widehat{\phi}_M - \phi_{k,M}\|_{L^2_{p_U}} > s_{N,M}\right]$$

$$\geqslant s_{N,M}^2 \frac{1}{K+1} \sum_{k=0}^K \mathbb{E}_{X^1,\dots,X^M}\left[\mathbb{P}_{\phi_{k,M}}\left(\|\widehat{\phi}_M - \phi_{k,M}\|_{L^2_{p_U}} > s_{N,M}\Big| X^1,\dots,X^M\right)\right], \tag{C.14}$$

where the last inequality follows since the maximal value over the functions is no less than the average and since $\mathbb{P}(A) = \mathbb{E}[1_A] = \mathbb{E}_Z[\mathbb{E}[1_A|Z]] = \mathbb{E}[\mathbb{P}(A|Z)]$.

Next, we transform to bounds in the average probability of testing error of the $2s_{N,M}$-separated hypothesis functions. Define $k_{\mathrm{test}}$ as the minimum distance test:

$$k_{\mathrm{test}} = \arg\min_{k=0,\dots,K} \|\widehat{\phi}_M - \phi_{k,M}\|_{L^2_{p_U}}.$$

Since $\phi_{k_{\mathrm{test}},M}$ is the closest one, we have that $\|\widehat{\phi}_M - \phi_{k_{\mathrm{test}},M}\|_{L^2_{p_U}} \leqslant \|\widehat{\phi}_M - \phi_{k,M}\|_{L^2_{p_U}}$ for all $k \neq k_{\mathrm{test}}$. Using the fact that the function $\phi_{k,M}$ are built as $2s_{N,M}$ separated functions and using the triangle inequality we have:

$$2s_{N,M} \leqslant \|\phi_{k,M} - \phi_{k_{\mathrm{test}},M}\|_{L^2_{p_U}} \leqslant \|\widehat{\phi}_M - \phi_{k_{\mathrm{test}},M}\|_{L^2_{p_U}} + \|\widehat{\phi}_M - \phi_{k,M}\|_{L^2_{p_U}} \leqslant 2\|\widehat{\phi}_M - \phi_{k,M}\|_{L^2_{p_U}},$$

so $s_{N,M} \leqslant \|\widehat{\phi}_M - \phi_{k,M}\|_{L^2_{p_U}}$ for all $k \neq k_{test}$. Hence,

$$\mathbb{P}_{\phi_{k,M}}\left(\|\widehat{\phi}_M - \phi_{k,M}\|_{L^2_{p_U}} \geqslant s_{N,M}\Big| X^1,\cdots,X^M\right) \geqslant \mathbb{P}\left(k_{\mathrm{test}} \neq k\big| X^1,\cdots,X^M\right). \tag{C.15}$$

Consequently,

$$\frac{1}{K+1} \sum_{k=0}^K \mathbb{P}_{\phi_{k,M}}\big(\|\widehat{\phi}_M - \phi_{k,M}\|_{L^2_{p_U}} \geqslant s_{N,M} \mid X^1,\cdots,X^M\big)$$

$$\geqslant \inf_{k_{\mathrm{test}}} \frac{1}{K+1} \sum_{k=0}^K \mathbb{P}_{\phi_{k,M}}\big(k_{\mathrm{test}} \neq k \mid X^1,\cdots,X^M\big) = \inf_{k_{\mathrm{test}}} \frac{1}{K+1} \sum_{k=0}^K \bar{\mathbb{P}}_k\big(k_{\mathrm{test}} \neq k\big) \tag{C.16}$$

where $\bar{\mathbb{P}}_k(\cdot) = \mathbb{P}_{\phi_{k,M}}(\cdot \mid X^1,\dots,X^M)$.

The Kullback divergence estimate in equation (D3) from Lemma 4.2 holds with $0 < \alpha < 1/8$, and by Lemma C.2 and the fact that $K = 2^{\lceil c_{0,N} M^{\frac{1}{2\beta+1}}\rceil}$ in equation 4.3 increases exponentially in $M$, we have:

$$\inf_{k_{\mathrm{test}}} \frac{1}{K+1} \sum_{k=0}^K \bar{\mathbb{P}}_k\big(k_{\mathrm{test}} \neq k\big) \geqslant \frac{\log(K+1) - \log(2)}{\log(K)} - \alpha \geqslant \frac{1}{2} \tag{C.17}$$

if $M$ is large. Note that the above lower bound of $\inf_{k_{\text{test}}} \frac{1}{K+1} \sum_{k=0}^{K} \bar{\mathbb{P}}_k \left( k_{\text{test}} \neq k \right)$ is independent of the dataset $\{X^m\}_{m=1}^M$. Using equation C.17 ,equation C.16 and equation C.14, we obtain with $c_0 = \frac{1}{2}[C_1 C_0^{-\beta}]^2$,

$$\sup_{\phi_\star \in \mathcal{C}^s(L,\bar{a})} \mathbb{E}_{\phi_\star} \left[ \|\widehat{\phi}_M - \phi_\star\|_{L^2_{p_U}}^2 \right] \geqslant \frac{s_{N,M}^2}{2} = c_0 N^{-\frac{2\beta}{2\beta+1}} M^{-\frac{2\beta}{2\beta+1}} \tag{C.18}$$

for any estimator. Hence, the lower bound equation 4.3 holds. □

**Proof of Theorem 4.4** First, we reduce the supremum over all $A_\star$ to a single one. Let $A^1 \in \mathcal{A}_d(r,\bar{a})$ with $\text{rank}(A^1) \geqslant 2$. Since

$$\mathcal{G}_{A^1} := \left\{ g_{\phi,A^1}(x,y) = \phi(x^\top A^1 y) \ : \phi \in \mathcal{C}^\beta(L,\bar{a}), \ \|\phi\|_\infty \leqslant B_\phi \right\} \subseteq \mathcal{G}_r^\beta(L, B_\phi, \bar{a}),$$

we have for any $\widehat{g}$,

$$\sup_{g_\star \in \mathcal{G}_r^\beta(L,B_\phi,\bar{a})} \mathbb{E}\|\widehat{g} - g_\star\|_{L^2_\rho}^2 \geqslant \sup_{g_\star \in \mathcal{G}_{A^1}} \mathbb{E}\|\widehat{g} - g_\star\|_{L^2_\rho}^2. \tag{C.19}$$

Thus, to prove equation 4.5, it suffices to prove it with $g_\star \in \mathcal{G}_{A^1}$.

Let $U^1$ be the random variable defined in (4.1) with $A_\star = A^1$. Then, Lemma 4.1 implies that

$$\|\widehat{g} - g_\star\|_{L^2_\rho}^2 \geqslant \|\hat{\psi} - \phi_\star\|_{L^2_{p_{U^1}}}^2$$

for any $\widehat{g}(x,y) := \widehat{\phi}(x^\top \widehat{A} y)$ with $\widehat{\phi} \in L^2_{p_{U^1}}$ and $\widehat{A} \in \mathcal{A}_d(r,\bar{a})$ and any $g_\star \in \mathcal{G}_{A^1}$. Here, $\hat{\psi}$, defined in equation 4.2, varies according to $\widehat{g}$ since both $A_\star$ and the distribution of $X$ are fixed. Taking first the expectation over $\widehat{g}$, then taking the supremum over $g_\star \in \mathcal{G}_{A^1}$ followed by the infimum over $\widehat{A}$ and $\widehat{\phi}$, we obtain

$$\inf_{\substack{\widehat{A} \in \mathcal{A}_d(r,\bar{a}) \\ \widehat{\phi} \in L^2_{p_{U^1}}}} \sup_{g_\star \in \mathcal{G}_{A^1}} \mathbb{E}\|\widehat{g} - g_\star\|_{L^2_\rho}^2 \geqslant \inf_{\hat{\psi} \in L^2_{p_{U^1}}} \sup_{\substack{\phi_\star \in \mathcal{C}^\beta(L,\bar{a}) \\ \|\phi_\star\|_\infty \leqslant B_\phi}} \mathbb{E}\|\hat{\psi} - \phi_\star\|_{L^2_{p_{U^1}}}^2. \tag{C.20}$$

Meanwhile, Theorem 4.3 gives a lower bound

$$\liminf_{M \to \infty} \inf_{\hat{\psi} \in L^2_{p_{U^1}}} \sup_{\substack{\phi_\star \in \mathcal{C}^\beta(L,\bar{a}) \\ \|\phi_\star\|_\infty \leqslant B_\phi}} M^{\frac{2\beta}{2\beta+1}} \mathbb{E}\|\hat{\psi} - \phi_\star\|_{L^2_{p_{U^1}}}^2 \geqslant c_0 N^{-\frac{2\beta}{2\beta+1}} \tag{C.21}$$

with $c_0 > 0$.

Combining (C.19)–(C.21), we then obtain:

$$\liminf_{M \to \infty} \inf_{\widehat{g}} \sup_{g_\star \in \mathcal{G}_r^\beta(L,B_\phi,\bar{a})} M^{\frac{2\beta}{2\beta+1}} \mathbb{E}\left[ \|\widehat{g} - g_\star\|_{L^2_\rho}^2 \right]$$

$$\geqslant \liminf_{M \to \infty} \inf_{\widehat{g}} \sup_{g_\star \in \mathcal{G}_{A^1}} M^{\frac{2\beta}{2\beta+1}} \mathbb{E}\left[ \|\widehat{g} - g_\star\|_{L^2_\rho}^2 \right]$$

$$\geqslant \liminf_{M \to \infty} \inf_{\hat{\psi} \in L^2_{p_{U^1}}} \sup_{\substack{\phi_\star \in \mathcal{C}^\beta(L,\bar{a}) \\ \|\phi_\star\|_\infty \leqslant B_\phi}} M^{\frac{2\beta}{2\beta+1}} \mathbb{E}\|\hat{\psi} - \phi_\star\|_{L^2_{p_{U^1}}}^2 \geqslant c_0 N^{-\frac{2\beta}{2\beta+1}},$$

which gives the desired result in equation 4.5. □

## D  NUMERICAL SIMULATIONS CONFIGURATION

This section provides a detailed description of the simulations presented in Section 5.

### D.1  DATA GENERATION

For each sample size $M$, we run a Monte Carlo simulation over different seeds as follows. We draw token arrays $X^{(m)} = (X_1^{(m)}, \ldots, X_N^{(m)}) \in \mathcal{C}_d^N$ i.i.d. with $X_i^{(m)} \sim \text{Unif}[0,1]^d/\sqrt{d}$ sampled i.i.d

and construct the $(X_i^{(m)})^\top A_\star X_j^{(m)}$ terms, evaluate the interaction via $\phi_\star$ (the sampling method of $\phi_\star$ and $A_\star$ is detailed below, and aggregate and add i.i.d. noise $\eta_i^{(m)} \sim \mathcal{N}(0, \sigma^2)$ as described in equation 2.1 to generate $Y_i^{(m)}$.

For each simulation, we sample the ground truth interaction $g_\star(x, y) = \phi_\star(x^\top A_\star y)$ by drawing random $\phi_\star$ and choosing $A_\star$. We represent $\phi_\star$ as a B-spline of degree $P_\star$ defined on an open uniform knots with $K$ basis functions on $[-1, 1]$.

$$\phi_\star(u) = \sum_{k=1}^{K_\star} \theta_\star^k B_k(u).$$

For each seed, we draw $\theta_\star \sim \mathcal{N}(0, I_{K_\star})$ and then normalize it for $\|\theta_\star\| = \sqrt{K_\star}$.

## D.2 ESTIMATOR

If $A_\star$ was known, the estimator can be computed by setting $\hat{A} = A_\star$ and setting $\hat{\phi}(u) = \sum_{k=1}^K \hat{\theta}_k B_k(u)$ with degree $P_{\text{est}}$ and $\hat{\theta}$ chosen according to the ridge regression formula:

$$\hat{\theta} = \left(U^\top U + \lambda_\theta I\right)^{-1} U^\top y, \tag{D.1}$$

where $U = \left(U_{(m,i),k}\right) \in \mathbb{R}^{MN \times K}$ with

$$U_{(m,i),k} := \frac{1}{N-1} \sum_{j \neq i} B_k\left((X_i^{(m)})^\top A X_j^{(m)}\right) \tag{D.2}$$

and $y = \left(Y_i^{(m)}\right) \in \mathbb{R}^{MN \times 1}$.

However, since $A_\star$ is unknown, the joint estimation of $(A, \phi)$ is non-convex due to the composition $\phi(x^\top A y)$. To mitigate local minima, we use a hot start and an alternating scheme. We perform the hot start by setting $A^{(0)} = A_\star + \Delta_A$ with $\Delta_A$ being a perturbation specified in Table 1 and setting the initial $\theta^{(0)}$ as the matching ridge solution D.1. In the PyTorch implementation, the scheme includes a description of the function $\hat{\phi}$ as a neural network. This is because representing it as B-splines directly would require differentiating through the B-spline basis, which is cumbersome for automatic differentiation. To address this, we introduce a neural-network surrogate $\Phi_{\text{net}}$ that approximates the spline and can be used as a differentiable link in the $A$-step.

**Alternating Optimization for $\hat{\phi}, \hat{A}$**

1. **Hot start:** set $A^{(0)} = A_\star + \Delta_A$ and compute $\theta^{(0)} = \left(U^\top U + \lambda_\theta I\right)^{-1} U^\top y$ with $U$ computed according to $A^{(0)}$ in equation D.2

2. **For $t = 1, \ldots, T$:**

   (a) *Approximate the current spline using a multilayer perceptron (MLP).* Fit an MLP $\Phi_{\text{net}}^{(t-1)}$ on a grid $\{u_\ell\}$ to minimize $\sum_\ell |\Phi_{\text{net}}^{(t-1)}(u_\ell) - \tilde{\phi}^{(t-1)}(u_\ell)|^2$

   (b) *A-step through optimization.* Update $A$ by minimizing the empirical loss with $\hat{\phi} = \Phi_{\text{net}}^{(t-1)}$ held fixed using the Adam optimizer

   $$\min_A \frac{1}{MN} \sum_{m=1}^M \sum_{i=1}^N \left(\sum_{j \neq i} \hat{g}_{A, \Phi_{\text{net}}^{(t-1)}}(X_i^{(m)}, X_j^{(m)}) - Y_i^{(m)}\right)^2 + \frac{\lambda_A}{2} \|A\|_F^2. \tag{D.3}$$

   (c) *$\theta$-step through closed form.* With $A$ fixed at $A^{(t)}$, compute $\theta^{(t)}$ by ridge regression: stack $y \in \mathbb{R}^{MN}$ from $Y_i^{(m)}$, and build $U \in \mathbb{R}^{MN \times K}$ with

   $$U_{(m,i),k} = \frac{1}{N-1} \sum_{j \neq i} B_k\left((X_i^{(m)})^\top A^{(t)} X_j^{(m)}\right)$$

   and compute

   $$\theta^{(t)} = \left(U^\top U + \lambda_\theta I\right)^{-1} U^\top y.$$

**Choice of $K_{\text{est}}$ and $\lambda_\theta$.** We set the number of spline coefficients by the bias variance trade-off for a $\beta$-Hölder smoothness as done in equation B.20

$$K_{\text{est}} = \text{round}\left(K_{\text{scale}}(M/\log M)^{1/(2\beta+1)}\right)$$

where $K_{\text{scale}}$ is a chosen constant. For the ridge regularization constant $\lambda_\theta$ we follow the standard scaling for least squares models with $MN$ responses and $K$ coefficients, the variance of $\hat{\theta}$ should scale like $K/(MN)$, so we take

$$\lambda_\theta = \lambda_{\text{scale}} \frac{K_{\text{est}}}{M(N-1)},$$

## D.3 Error estimate

We measure accuracy via the estimator test MSE, sampling never seen inputs $X^{(m)} \sim \text{Unif}[0,1]^d/\sqrt{d}$ and evaluating:

$$\text{MSE}_g = \frac{1}{N_{\text{test}}} \sum_{m=1}^{N_{\text{test}}} \frac{1}{N(N-1)} \sum_{i=1}^{N} |\sum_{j \neq i} \hat{g}_{\hat{A},\hat{\phi}}(X_i^{(m)}, X_j^{(m)}) - g_\star(X_i^{(m)}, X_j^{(m)})|^2.$$

## D.4 Simulation parameters

The following table details the parameters used for the simulations described in Section 5.

Table 1: Chosen parameters for the simulation

| Parameter | Value |
| --- | --- |
| Seeds | 300 |
| $A_\star$ | Diagonal matrix with i.i.d. entries $A_{11} = 1$, $\forall i > 1$ $A_{ii} \sim \text{Unif}[-1,1]$ |
| Sample sizes $M$ | [20000, 27355, 37416, 51177, 70000] |
| $N$ | 3 |
| Gaussian noise std $\sigma_\eta$ | 0.07 (Gaussian) |
| Estimator degree | $P_{\text{est}} = P_\star$ |
| $K_\star$ | 16 |
| $K_{\text{scale}}$ | [(a) and (b) for $P_\star = 3$]: 16 |
| | [(b) for $P_\star = 8$]: 30 |
| Basis size $K_{\text{est}}$ | [(a) and (b) for $P_\star = 3$]: $\{73, 78, 82, 87, 92\}$ (matching the $M$ grid) |
| | [(b) for $P_\star = 8$]: $\{50, 51, 52, 53, 54\}$ (matching the $M$ grid) |
| $\lambda_A$ | $10^{-5}$ |
| $\lambda_{\text{scale}}$ | 2 |
| $\lambda_\theta$ | [(a) and (b) for $P_\star = 3$] $10^{-3} \times \{6.85, 5.30, 4.12, 3.19, 2.46\}$ |
| | [(b) for $P_\star = 8$]$\{2.50, 1.86, 1.39, 1.04, 0.77\}$ |
| | (matching the $M$ grid) |
| $\Delta_A$ | Entry wise Gaussian noise with an std of $5/d \times 10^{-7}$ |
| $T$ | 4 |
| $A$-step optimizer | Adam, lr $= 10^{-8}$, 20 epochs. |
| $\Phi_{\text{net}}^{(t)}$ architecture | 1-hidden layer of width 32 (GELU activation) |
| $\Phi_{\text{net}}^{(t)}$ optimization | 1000 epochs (Adam) lr = 0.01 |
| Test set | 2000 samples |

