# OpenReview forum: "Minimax Rates for Learning Pairwise Interactions in Attention-Style Models"
_ICLR.cc/2026/Conference — ICLR 2026 Poster_

### Official Review · Reviewer_Yn8U · 2025-10-16

**Soundness:** 4
**Presentation:** 3
**Contribution:** 4
**Rating:** 10
**Confidence:** 4

**Summary:**

This paper studies the statistical problem of learning pairwise interactions in sequences with attention layers. The learning task is a sequence-to-sequence task, which consists in an attention matrix (i.e., a quadratic form in the input tokens) followed by element-wise activation and averaging. The goal is to learn the activation function (among a class of $\beta$-smooth functions) and the attention matrix (which can incorporate a rank constraint). Authors show that the empirical risk minimizer reaches minimax optimality for this problem, and that the minimax rate depends only on the smoothness of the activation, and not on the dimension of the data and on the sequence length.

**Strengths:**

The paper tackles important questions related to characterizing the statistical properties of sequence-to-sequence tasks that are representative of settings where Transformer performs well (which typically differ from usual time series models in statistics). In particular the paper studies pairwise interactions between input tokens, which are known to be a crucial feature of attention models, under reasonable assumptions on the data distribution. Obtaining the minimax rate and showing that it is reached by the ERM is a good contribution. The paper is clear to follow, and the proofs seem correct.

**Weaknesses:**

I did not see any major weakness. While I give suggestions below to improve the impact of the paper, I do not expect the authors to answer on these points during the rebuttal.

Although the paper is well-written, I encourage the authors to revisit their paper to give more interpretation of what is going on. For instance, while the authors state several times that the attention avoids the curse of dimensionality in their model, they do not explain _why_: because the nonparametric estimation problem is a unidimensional problem, while in high-dimension you "only" have to estimate a matrix. Similarly, the authors begin in the introduction by stating that the problem is difficult because we only observe an average interaction over the tokens. I imagine this is solved by the exchangeability assumption, which also explains why the sequence length does not appear in the rate, but this could be explained a bit more clearly. I also think it would be very insightful to state the equation on line 859 as an intermediate result in the main text, before deriving the asymptotic rate by assuming $rd$ is small enough and solving for $K_M$.

Minor remarks:
- line 183: while I appreciate going from independence to exchangeability, I suggest rephrasing this paragraph, because exchangeability is not much more realistic than independence when thinking about NLP.
- line 238: $R_g[X]$ is not defined yet at this point.
- line 243: please explain somewhere in the main text how $K_M$ is chosen. Besides, $K_M$ is a natural number so there is a floor function missing in equation (A.20).
- line 252: state that this estimator is the empirical risk minimizer.
- line 292: I disagree with the formulation of the last sentence of the remark. It suggests that taking a rougher activation function helps to mitigate the curse of dimensionality, which does not seem to be a correct interpretation. Indeed, in the proof, before replacing $rd$ using the assumption of the theorem in eq. (A.19), one sees that increasing either $d$ or $\beta$ can only degrade the bound. I would rather say that the contribution of the dimension $d$ to the error comes from the estimation of $A_*$, and that this term does not dominate as long as $rd \leq (M/\log M)^{1/2\beta + 1}$, and that for a fixed $d$ this condition becomes looser as $\beta$ decreases because the other error terms increase.

**Questions:**

- line 125: most theoretical works rather assume that the particles are on the sphere rather than on the cube, due to the layer normalization operation in Transformers. Do you think it would be possible to change to the sphere?
- line 150: the formulation is misleading as it may lead the reader to believe that softmax is included in the setting under study, whereas only element-wise activations are included. How difficult would it be to extend the results to a function $\phi_\star: \mathbb{R}^n \to \mathbb{R}$?

---

> ### Author Response · Authors · 2025-11-20
> **Response to Yn8U Review 1**
>
> We thank you for your constructive and detailed feedback, and are glad you found our paper insightful. We found your suggestions for improving the paper's clarity and impact very helpful, and we have fixed the typos and misspells you mentioned. In addition, we have added more interpretation based on your suggestions.
>
> # Response to comments and suggestions
>
> Following the reviewer's great comments and suggestions, we have made the following changes in the revision.
>  * Regarding the rate being dimension-free: You are exactly right. The problem's structure allows the error to be decomposed into a 1D non-parametric estimation of $\phi$ (with its characteristic rate $M^{-2\beta/(2\beta+1)}$) and a parametric estimation of the matrix $A$ (with rate $O(rd/M)$). Our dimension-free rate holds in the regime where the parametric error is dominated by the non-parametric error. We have added a clarification for that in the paper in the presentation of the result in the main contributions section.
>  * As the reviewer correctly stated, the problem is difficult because we only observe an average interaction over the tokens. And as the reviewer pointed out,  having exchangeability ensures coercivity, which makes the problem solvable. We added a clarification on that in the main text before Lemma 3.4. We note that for the lower bound, the exchangeability is not required.
>  * The Intermediate Result (Eq. 859): This is an excellent suggestion. In the revised version, we have Corollary B.2, which presents a more general bound without imposing the assumption $rd\leq \left(M/\log M \right)^{1/(2\beta + 1)}$. The corollary illustrates the nonparametric and parametric errors. Imposing the constraint on $rd$, the main result is obtained. We have also revised remark 3.3 and highlighted this intermediate result.
> # Response to minor remarks
>
> * **Exchangeability Assumption** we agree with the reviewer. We removed that sentence connecting to NLP. We have added a clarification of the assumption of exchangeability, also linking to relevant literature in lines 210-213 of the main text.
> * Regarding line 238, thank you for the accurate catch; we've fixed those mistakes in the revised version.
> * Thank you for the correction. We have added the floor function. We also added a comment that "This choice of $K_M$ balances the terms in Eq. B.19 and leads to a tighter upper bound."
> * Thank you for the correction. We have added that the estimator is the empirical risk minimizer at the beginning of Section 3.
> * Regarding the remark on line 292: thank you for the correction. We have updated this remark on the revised manuscript, as you correctly suggested, focusing on parametric term controlling the bound when $\beta$ gets smaller.
> # Response Regarding the Questions
>
> * Yes, it is possible to generalize the minimax rates to particles on the sphere. Both the upper and lower bounds can be extended with largely the same proofs, except subtle technical changes related to the probability densities on the sphere. For the upper bound, the exploration measure $\rho$ now becomes a measure on $\mathbb{S}^{d-1}\times \mathbb{S}^{d-1}$, which is still a compact set. For the lower bound, we need to prove that the probability densities of the random variables $U_{ij}= X_i^\top A X_j$ exist and are continuous. In the current proof, we use the fact that the joint density of $(X_i,X_j)$ exists in $\mathbb R^{2d}$ and is continuous on the cube, so that the existence of the density of $U_{ij}$ follows by a change of variable and SVD of $A$. On the sphere, we can assume that the joint density of $(X_i,X_j)$ exists on the manifold $\mathbb{S}^{d-1}\times \mathbb{S}^{d-1}$, and use a local chart to perform the change of variable. Thus, the generalization is possible with subtle technical changes.

---

> > ### Author Response · Authors · 2025-11-20
> > **Response to Yn8U Review 2**
> >
> > * We thank the reviewer for pointing out the imprecision in Line 150. We agree that the original phrasing could be interpreted as claiming our model captures the global normalization of the Softmax mechanism, whereas our model uses an averaging normalization and only includes element-wise (pairwise) activations. Specifically, our model simplifies the normalization factor $Z_i$ in $Y_i=\sum_{j=1}^{N} \frac{1}{Z_i}g(X_i,X_j) V_j$ from $Z_i:=\sum_{l=1}^{N} g(X_i,X_l)$ with $g(x,y)= c e^{\beta x^\top A y}$ in Softmax mechanism to simply $Z_i\equiv Nc$. Such a simplificiation relies on a mean-field approximation: as the number of tokens (particles) $N$ becomes large, the partition function (the Softmax denominator) concentrates around its mean. This allows us to treat the normalization as a constant absorbed into the nonlinearity, thereby reducing the global Softmax operation to the pairwise interaction form studied in our paper. Also, this simplification allows us to isolate the core learning problem of estimating the nonlocal interaction function $g(x,y)= \phi(x^\top A y)$, since the nonlocal dependence is a key feature of the attention mechanism.
> >   We have added a new section, Appendix A: Reduction from Attention to IPS Attention Model, which formally details this reduction. It cites established works [1,2] that utilize similar simplifications to bridge the gap between standard Self-Attention and Interacting Particle Systems (IPS).
> > * **Regarding extending to a general $\phi:\mathbb R^N\rightarrow\mathbb R$** Extending the theoretical results to a general function $\phi:\mathbb R^N \to \mathbb R$, e.g., in the form of $\phi(X_i^\top A X_1,\cdots, X_i^\top A X_N )$, would be significantly more difficult. The main challenge lies in the high dimensionality of the input space for $\phi$, which would lead to a curse of dimensionality in estimating $\phi$ non-parametrically.
> >   This type of function class is too large and therefore produces a looser bound. Since the dependence of all particles/tokens in the softmax is incorporated through normalization, we feel this will not account for the problem's symmetries. In particular, we believe the pairwise formulation captures the softmax attention mechanism's essential "nonlocal interaction" in the large-sequence limit (see also Appendix A).
> >   A possible extension that may account for the symmetries of the problem and include the softmax normalization is to consider a model of the form, $Y_i = \frac{1}{N} \sum_{j=1}^N \phi_i(X_i ^\top A_\star X_j) +\eta_i$, as this will account for the softmax normalization. However, this will not change the rate. Our focus here was on the simplest model, which illustrates the transformer's dimension-free performance.
> > 1) Michael E. Sander, Pierre Ablin, Mathieu Blondel, and Gabriel Peyré. Sinkformers: Transformers with doubly stochastic attention. In Gustau Camps-Valls, Francisco J. R. Ruiz, and Isabel Valera (eds.), Proceedings of The 25th International Conference on Artificial Intelligence and Statistics, volume 151 of Proceedings of Machine Learning Research, pp. 3515–3530. PMLR, 28–30 Mar 2022.
> > 2) Borjan Geshkovski, Cyril Letrouit, Yury Polyanskiy, and Philippe Rigollet. A mathematical perspective on transformers. Bulletin of the American Mathematical Society, 62(3):427–479, 2025.

---

> > > ### Comment · Reviewer_Yn8U · 2025-11-25
> > >
> > > Thank you for the rebuttal that adequately addresses my questions. I keep my score.

---

### Official Review · Reviewer_o395 · 2025-10-29

**Soundness:** 3
**Presentation:** 3
**Contribution:** 2
**Rating:** 4
**Confidence:** 2

**Summary:**

The paper studies a model of $N$ particles with pairwise interactions which resembles attention networks. In particular, one can consider the problem under study as being given $M$ examples of $N$ tokens each in dimension $d$ and the measures of their interaction strength, defined as an element-wise non-linearity applied to a linear dot-product attention of rank $r$ (plus some noise). Given this data, you wish to recover both the attention matrix and the non-linearity. The authors prove that the error on learning the interaction models decays as a certain dimension-free power of the number of examples $M$ as long as the product $rd$ is smaller than a power of $M/log(M)$. This result is verified in one case on synthetic data.

**Strengths:**

The paper is clear and well written, making the proof easy to follow. Within the modeled under consideration the results are intriguing and sufficiently novel. In particular the authors have very permissive assumptions on the distribution of the examples and the additive noise. I believe the contribution to the theory landscape can be significant, as not many theory papers tackle the problem of dependent token sequences.

**Weaknesses:**

Although the mathematical analysis is solid, the link to real attention mechanisms is tenuous. The model studied is not an actual dot-product attention layer: it lacks the row-wise softmax normalization that defines attention, and instead applies an element-wise nonlinearity to pairwise scores. As a result, the interacting-particle formulation captures only a simplified kernel-like approximation of attention. I would have liked the authors to better justify why this simplified model is a meaningful generative proxy for attention data, and why its minimax rates are relevant to understanding transformers. A broader empirical study comparing to actual softmax attention or multi-head layers could have clarified the connection and strengthened the practical implications.

Minor typo: in line 154 you refer to $g_*$ in 2.1, which is not there, but a few lines later.

**Questions:**

1. Would it be possible to extend your model to include row-wise non-linearities? I believe it would be incredibly interesting, as it would be a data generating model that naturally mimics softmax attention.
2. Could you provide some practical examples of data that you believe can be effectively described by your particle model?

---

> ### Author Response · Authors · 2025-11-20
> **Response to o395 Review 1**
>
> Thank you for your thoughtful review and comments. We are glad that you found our paper clear and mathematically sound. We appreciate your comments on the link to the attention mechanism, which have helped us improve the paper, particularly on clarifying the connection between our model and the self-attention mechanism and our simplification in row-wise normalization.
>
> # Response Regarding Weaknesses
>
> * **Simplification of row normalization to an average** We appreciate your comment regarding the simplification of the row-wise softmax normalization to an average, i.e., simplifying the normalization factor $Z_i$ in $Y_i=\sum_{j=1}^{N} \frac{1}{Z_i} g(X_i,X_j) V_j$ from $Z_i:=\sum_{l=1}^{N} g(X_i,X_l)$ with $g(x,y)= c e^{\beta x^\top A y}$ to $Z_i\equiv Nc$, and we also remove the value $V_j=W_V X_j$ to further simplify the problem. This simplification was made to isolate the core learning problem of estimating the nonlocal interaction function $g(x,y)= \phi(x^\top A y)$, since the nonlocal dependence is a key feature of the attention mechanism.
>   We have added a section in the revised manuscript (Appendix A) that provides a detailed connection between our model and the self-attention mechanism. In particular, we note that when $N$ is sufficiently large, $Z_i=\sum_{l=1}^{N} g(X_i,X_l)\approx N \mathbb{E}[g(X_i,X_l) \mid X_i]$, so the normalization in the softmax can be approximated by its mean value, leading to a model similar to ours. This reduction is inspired by prior work [1], which also employs a similar simplification to capture the qualitative behavior of self-attention.
> * **Minimax rates relevance to transformers** Minimax rates provide fundamental limits on the sample complexity of learning models, which is crucial for understanding the efficiency of learning algorithms. In the context of transformers, our results shed light on how many samples are needed to accurately learn the interaction structure defined by the kernel function $\phi$ and the attention matrix $A$ in a simplified attention-type model. This understanding in a simplified model is the first step in further understanding the sample complexity of transformers. That said, we acknowledge the importance of the value matrix and multi-head attention, and we are currently working on extending our analysis to these models.
>
> # Response Regarding Questions
>
> * Thank you for your question. Does the reviewer mean by row-wise function of the form $\phi(X_i^\top A X)$ with $X\in \mathbb R^{d\times N }$ and $X_i\in \mathbb R^d$? If so, this type of function class is too large and will produce a looser bound. Since the dependence of all particles/tokens in the softmax is incorporated through normalization, we feel this will not account for the problem's symmetries (see also Appendix A). A possible extension that may account for the symmetries of the problem and include the softmax normalization is to consider a model of the form, $Y_i = \frac{1}{N} \sum_{j=1}^N \phi_i(X_i ^\top A_\star X_j) +\eta_i$, as this will account for the softmax normalization. However, this will not change the rate. Learning a model that includes the value matrix together with $\phi$ and $A$ is a more challenging and exciting future direction we are currently developing. Our focus here was on the simplest model, which illustrates the dimension-free transformer. See also our reply to reviewer Yn8u.

---

> > ### Author Response · Authors · 2025-11-20
> > **Response to o395 Review 2**
> >
> > * **Regarding the Practical Examples of the Model** Thank you for your question. Our particle system view applies directly to attention layers for which the softmax is replaced by an element‑wise activation or a kernel feature map. In such cases, the output can be presented directly as a sum over interactions. There are several methods that use an attention mechanism with other activation to reduce computational complexity and, in some cases, improve performance.  We added a section in the related work with reference to some of these works.
> > 	 *  Activation-based Transformers: Our model directly describes attention models that replace the softmax with element-wise activations. We have detailed in the related works examples of such use cases, including the linear attention [2] and the ReLU-based model [3].
> > 	*  As for vision tasks, several Vision Transformer (ViT) variants remove the softmax activation while remaining competitive. For example,[4] consider an attention mechanism based on a Gaussian kernel, and [5] apply linear attention after normalizing the Key-Query columns. Furthermore, [6] examines a sigmoid function as the attention activation, showing it acts as a universal function approximator and benefits from improved regularity compared to softmax attention.
> > 1) Geshkovski, B., Letrouit, C., Polyanskiy, Y., & Rigollet, P. (2025). A mathematical perspective on transformers. Bulletin of the American Mathematical Society, 62(3), 427-479.
> > 2) Von Oswald, J., Niklasson, E., Randazzo, E., Sacramento, J., Mordvintsev, A., Zhmoginov, A. & amp; Vladymyrov, M.. (2023). Transformers Learn In-Context by Gradient Descent. Proceedings of the 40th International Conference on Machine Learning, in Proceedings of Machine Learning Research  202:35151-35174
> > 3) Biao Zhang, Ivan Titov, and Rico Sennrich. Sparse attention with linear units. In Marie-Francine
> >     Moens, Xuanjing Huang, Lucia Specia, and Scott Wen-tau Yih (eds.), Proceedings of the
> >     2021 Conference on Empirical Methods in Natural Language Processing, EMNLP 2021, Vir-
> >     tual Event / Punta Cana, Dominican Republic, 7-11 November, 2021, pp. 6507–6520. Associ-
> >     ation for Computational Linguistics, 2021.
> > 4) Jiachen Lu, Jinghan Yao, Junge Zhang, Xiatian Zhu, Hang Xu, Weiguo Gao, Chunjing Xu, Tao Xiang, and Li Zhang. Soft: Softmax-free transformer with linear complexity. In NeurIPS, 2021.
> > 5) Soroush Abbasi Koohpayegani and Hamed Pirsiavash. Sima: Simple softmax-free attention for vision transformers. In 2024 IEEE/CVF Winter Conference on Applications of Computer        Vision (WACV), pp. 2595–2605, 2024.
> > 6) Jason Ramapuram, Federico Danieli, Eeshan Dhekane, Floris Weers, Dan Busbridge, Pierre Ablin, Tatiana Likhomanenko, Jagrit Digani, Zijin Gu, Amitis Shidani, and Russ Webb. Theory, analysis, and best practices for sigmoid self-attention. In International Conference on Learning Representations (ICLR), 2025.

---

### Official Review · Reviewer_QFiw · 2025-10-31

**Soundness:** 3
**Presentation:** 3
**Contribution:** 1
**Rating:** 2
**Confidence:** 4

**Summary:**

The paper studies learning interactions in attention-style models where the parameters are unknown. Specifically, they consider the function $Y_i=\frac{1}{N-1}\sum_{j\ne i} \phi(X_i^\top AX_j) +\text{noise}$ from $M$ i.i.d. samples $(X,Y)$ where input $X=(X_1,\cdots,X_N)$ consists of $N$ possibly dependent $d$-dimensional tokens and output $Y=(Y_1,\cdots,Y_N)$ measures average interactions. When $A$ is a rank $r$ matrix and $\phi$ is a $\beta$-Hölder function, the empirical risk minimizer is shown to have error $M^{-\frac{2\beta}{2\beta+1}}$ when $M\gtrsim (rd)^{2\beta+1}$.

**Strengths:**

The paper derives near-optimal minimax rates for estimation of interaction models such as a self-attention layer. This goes beyond existing frameworks for analyzing optimality of regression models as each sample consists of a possibly dependent sequence of tokens.

**Weaknesses:**

My main concern is novelty and motivation. I do not believe the results are interesting enough to merit publication.

* The problem is not clearly motivated. When do we need to estimate unknown parameters of attention models?

* The main result (i.e. the dimension-free rate) seems straightforward. Since $\phi$ is a 1-dimensional $\beta$-Hölder function (not $d$-dimensional), it is intuitive that $d$ does not appear in the exponent, and so the dependency on $d$ only arises through learning $A$ which is $rd$-dimensional due to low rank. So there is no "curse of dimensionality" to avoid in the first place, this does not really point to a "fundamental statistical efficiency" of attention models as the paper claims. Also for the comment in the abstract on weight matrix and activation not being separately identifiable, this is not really surprising as identifiability of parameters does not affect minimax rates in general.

* The paper claims the main challenge is going beyond the independency assumption, however the sample objects $X^m$ for $m=1,\cdots,M$ are assumed to be i.i.d. and only the tokens $X_1^m,\cdots,X_N^m$ can be dependent (with an exchangability assumption which immediately implies coercivity). Since the rate is given only for $M$ and not convergent in $N$, this also does not seem surprising. Other aspects of the proof seem more or less standard in minimax analysis (covering number bound, controlling tails by chaining, lower bound via Fano's inequality).

**Questions:**

See Weaknesses.

---

> ### Author Response · Authors · 2025-11-20
> **Response to QFiw Review**
>
> We thank you for your time and feedback. We appreciate that you found that our analysis of the sequence-dependent regression goes beyond the existing framework.
> We would like to respectfully clarify our perspective on the contribution and motivation, which we believe is significant.
>
> # Response Regarding Weaknesses
>
> * First, regarding your concern about motivation, the problem of "estimating unknown parameters of attention models" is the very definition of training a Transformer. Our work provides fundamental (minimax) limits on how many samples ($M$) are required to do so for this class of models, which we believe is a central and well-motivated question in the theory of deep learning.
> * On novelty and the "Curse of Dimensionality". You state that the dimension-free rate is "straightforward" because $\phi$ is a 1D function. We completely agree with your intuition, and this line of thought was our first step when constructing the theory behind the work. However, we don't agree that this intuition is commonly known or even that straightforward, as noted also by the other reviewers. Moreover, our contribution is centralized in rigorously proving that the complexity you find intuitive actually holds, and to find the constraints and cases under which this complexity holds, i.e., under the condition $M \gtrsim (rd)^{2\beta+1}$) which, at least in our view, is not trivial and intuitive.
> * The samples $M$ being independent is a standard assumption and is very mild. As noted by Reviewers Yn8u: "The paper is making a strong statement, which is interesting in many respects and raises a lot of questions. It seems technically solid (with quite general assumptions for the noise in particular)..." and xCfy: "In particular the paper studies pairwise interactions between input tokens, which are known to be a crucial feature of attention models, under reasonable assumptions on the data distribution". We note that the lower bound we prove via the Tsybakov method also yields the rate in $N$, namely $(MN)^{-\frac{2\beta}{2\beta+1}}$. In addition, it is for very general data distribution, which doesn't have to be exchangeable. The constraint we find on the number of particles in the upper bound, dimension, and smoothness, $M\gtrsim (rd)^{2\beta+1}$, is far from trivial and illustrates the role of the context length, smoothness, and dimension in the transformer.

---

### Official Review · Reviewer_xCfy · 2025-10-31

**Soundness:** 4
**Presentation:** 3
**Contribution:** 4
**Rating:** 8
**Confidence:** 2

**Summary:**

This paper concerns the study of asymptotic sample complexity for learning attention like models.
The main claim is that attention like models are free from the curse of dimensionality.
More precisely, while classical results for function regression leads to $M^{-2\beta/(2\beta+d)}$ scalings,
with beta the Holder exponent characterizing the smoothness of the function and d the embedding dimension,
they get a scaling corresponding to $d=1$.

**Strengths:**

The paper is making a strong statement, which is interesting in many respects and raises a lot of questions. It seems technically
solid (with quite general assumptions for the noise in particular), being grounded  on proven techniques (which I am not specialist of)
for obtaining minimax rates for non-parametric regressions,  which are adapted here to a context where the function is  observed through an average over particles, hence the function estimation is a deconvolution type inverse problem more involved than the usual single index model inference. The result is nicely backed by numerical  experiments. The paper is clearly written, despite the rather lengthy and technically complex  line of arguments.

**Weaknesses:**

Beside the technical proof, there is maybe a lack of interpretation of this result, which at least at first glance might look very surprising.
Not much intuition is given, and one wonder then if this quite paradoxical result stems from the assumptions which are taken.

**Questions:**

In order to better understand this result I have the following questions:

- do the scaling in $M^{-2\beta/(2\beta+1)}$ corresponds to what would be obtained if the attention matrix A would be known in advance,
because then we are back to the parallel regression of N functions of a real variable (x=X_i^tAX_j, leaving aside the extra difficulty of the convolution over the particles j)? If this interpretation is correct,  does this mean that the identification of A (i.e. the r directions among d)  do play a role only in a transient regime? is this the reason for the hypothesis that one need a number of samples $M > (rd)^{2\beta+1}$ (in my understanding this is the amount of samples it takes to approximately align with A), beyond which the sample complexity becomes 1d?
- Is this also related to the reason why the rank of the matrix is not involved in the asymptotic?
- I did not understand the role played by the interacting particle system interpretation for obtaining this result, it seems not to play any role in the proof.
- Concerning the regression assumption: usually the g function is actually not observed, it represents a  kind of virtual supervised function regression setting taking the functional form of an attention layer, so why should we expect this regression setting to be relevant to the full attention layer, i.e. when combining the attention weights with the values?
- if we compare to ordinary NN layers, should we obtain similar behaviour? I presume for a single layer function like $f(x) = W_1\phi(W_2^T x)$ the same rate should hold. What about more complex architectures? a rapid summary of existing results obtained so far for NN functions could be useful.

Overall my recomendation would be to save a little bit a room in the plain text to discuss the interpretation of the results.

---

> ### Author Response · Authors · 2025-11-20
> **Response to xCfy Review 1**
>
> # Response Regarding Weaknesses
>
> We thank the reviewer for the insightful comments, which helped us clarify the intuition behind the dimension-free rate, its relation to interacting particle systems, and the role of nonlocal dependence in the regression setting. These points have been incorporated into the revised manuscript.
>
> # Response Regarding Questions
>
> * **The scaling and dependence on attention matrix** Thank you for the insightful comment. Yes, you are correct, the scaling $M^{\frac{-2\beta}{2\beta + 1}}$ is indeed the optimal non-parametric rate for a 1D $\beta$-Hölder smooth function, or in essence, what would be obtained if the matrix $A_\star$ was known. You are also correct regarding the interpretation of the condition that $M > (rd)^{2\beta + 1}$. Mathematically, this condition emerges in the proof of the upper bound (Appendix A), controlling the "variance" term $T_{2,M}$, in step 5, combining the different error terms derived throughout the proof and observing the tradeoff from the terms stemming for the parametric and non parametric parts. For the non-parametric term to dominate, the parametric $O(\frac{rd}{M})$ term must decay at a faster rate, which in turn leads to this exact constraint.
>
> * Yes, as seen in the covering number argument, the rank mainly controls the effective dimension of the parameters, and as we consider values of $M$ that adhere to the condition, it also does not appear in the rate.
>
> * **Role of interacting particle systems (IPS) interpretation** Thank you for the comment. The IPS interpretation highlights the interaction structure of the attention model, in which the particles/tokens interact via a kernel function. This nonlocal interaction structure is central to both IPS and attention mechanisms. The IPS framework provides a natural way to model the aggregation of pairwise interactions, which is a key feature of attention layers. In our analysis, the IPS perspective is particularly useful for understanding the statistical properties of the model, as it emphasizes the role of the kernel function in defining the interactions. Moreover, the coercivity condition, which ensures the well-posedness of the problem, is inspired by similar conditions in the study of IPS. While the IPS interpretation does not directly affect the proof structure once coercivity condition is satisfied, it provides valuable intuition and a unifying perspective for the model under study.
>   Because of the strong similarity in structure between the attention layer and IPS, multiple works in the literature have drawn connections between the two, and there is a growing body of research exploring this relationship. The following references provide representative examples.
>     1) Geshkovski, B., Letrouit, C., Polyanskiy, Y., \& Rigollet, P. (2025). A mathematical perspective on transformers. Bulletin of the American Mathematical Society, 62(3), 427-479.
>     2) Geshkovski, Borjan, et al. "The emergence of clusters in self-attention dynamics." Advances in Neural Information Processing Systems 36 (2023): 57026-57037.
>     3) Dutta, Subhabrata, et al. "Redesigning the transformer architecture with insights from multi-particle dynamical systems." Advances in Neural Information Processing Systems 34 (2021): 5531-5544.

---

> > ### Author Response · Authors · 2025-11-20
> > **Response to xCfy Review 2**
> >
> > * **Regression assumption and relevance to full attention layer** We thank the reviewer for this comment, as it helps us to highlight a core feature of our model: the nonlocal dependence. As the reviewer pointed out, the function $g$ is not observed, and each output $Y_i = \frac{1}{N}\sum_{j=1}^Ng(X_i, X_j) +\eta_i$ depends on $g$ nonlocally through the average of multiple values, so the estimation of $g$ is like a deconvolution-type problem. This is different from the standard regression problem of estimating $f$ in $y = f(x) +\eta$, where each output $y$ depends on $f$ locally through a single value. This nonlocal dependence is a key feature of attention layers, where each token's output depends on the interactions with all other tokens. Therefore, our regression setting captures this essential aspect of attention mechanisms.  In this study, we focus on learning the interaction structure defined by the kernel function $\phi$ and the attention matrix $A_\star$, while ignoring the value matrix. This simplification allows us to isolate and analyze the statistically challenging aspect of learning the nonlocal interactions, without the effects of the value matrix.
> >   To clarify the connection between our model and the self-attention layer, we have added a section in the paper (Appendix A) that provides a direct connection and intuition for the model we study. In essence, we start from the softmax self-attention layer and show that when the number of tokens $N$ is sufficiently large we can replace the normalization of the softmax by an average, leading to a simplified model as ours.
> >   When the value matrix is included, the model becomes more complex, as each token's output is influenced by both the attention weights and the values. This introduces additional layers of complexity in the learning process, as one must now estimate both the interaction function and the value weight matrix. We totally agree with the reviewer that this is an important (and challenging) direction for future research, which we are currently pursuing. However, we believe that this study lays the foundation by addressing the core challenge of learning the nonlocal interaction.
> >
> >
> >  * **Related Works on Nonparametric and Semi Parametric works for Neural Networks** Thank you for your comment, we have added more related literature in the Related Works section of the updated version. Your intuition is correct, for a multi-index model $W_2\phi(W_1^Tx)$ with $W_2$ fixed, where $\phi$ acts element-wise, it was already shown by [1,2] that the rate remains $M^{-\frac{2\beta}{2\beta+1}}$. There are results for multi-layers of composition $\beta$-Hölder  smooth functions, see for example [3]; However, unlike our model, they are fully nonparametric.
> >    Closer to deep learning, [4] analyze generalized additive models with nested $k$-times differentiable compositions, showing the rate is $M^{-\frac{2k}{2k+1}}$. [5] proves that connected deep ReLU networks achieve a near-optimal minimax rate (up to log factors) over a class of composed functions. In [6] they study  a nonparametric interaction model in high dimension settings and show sparsity assumptions and associated regularization are required in order to obtain optimal rates of convergence.
> > 	1)  Stéphane Gaïffas. Guillaume Lecué. "Optimal rates and adaptation in the single-index model using aggregation." Electron. J. Statist. 1 538 - 573, 2007.
> > 	2) Györfi, László, et al. A distribution-free theory of nonparametric regression. New York, NY: Springer New York, 2002.
> > 	3) Schmidt-Hieber, Johannes. "Nonparametric regression using deep neural networks with ReLU activation function." (2020): 1875-1897.
> > 	4) Joel L. Horowitz and Enno Mammen. Rate-optimal estimation for a general class of nonparametric regression models with unknown link functions. The Annals of Statistics, 35(6):2589 – 2619, 2007.
> > 	5) Johannes Schmidt-Hieber. Rejoinder: “Nonparametric regression using deep neural networks with ReLU activation function”. The Annals of Statistics, 48(4):1916 – 1921, 2020.
> > 	6)  Sohom Bhattacharya, Jianqing Fan, and Debarghya Mukherjee. Deep neural networks for nonparametric interaction models with diverging dimension. The Annals of Statistics, 52(6):2738 – 2766, 2024.

---

### Author Response · Authors · 2025-12-03
**General Response for All Reviewers and AC**

We thank the AC and all reviewers for their careful reading and constructive feedback. Below, we briefly summarize the main clarifications and changes in the revised manuscript; detailed point-by-point responses are provided separately for each reviewer.


**Interpretation of the dimension-free rate.** Following the remarks of reviewers xCfy and Yn8u, we clarified that the estimation error decomposes into a 1D nonparametric term for $\phi_\star$ with rate $M^{-2\beta/(2\beta+1)}$ and a parametric term for $A_\star$ of order $O(rd/M)$. The dimension-free rate holds in the regime where the nonparametric term dominates, which yields the condition $M \gtrsim (rd)^{2\beta+1}$. This decomposition is now highlighted as an intermediate bound (Corollary B.2).

**Connection to softmax attention and IPS model.** We now provide a clear motivation for our IPS attention model, including a direct connection to softmax attention, as requested by reviewers o395 and Yn8u. This is now presented in a new section called "Reduction from attention to IPS attention model". In that section, we showed that, when $N$ is large, the softmax normalization concentrates around its mean. Absorbing this factor into the nonlinearity and considering approximately constant value vectors leads to our IPS attention model
    $Y_i \approx \frac{1}{N}\sum_{j\neq i} \phi(X_i^\top A X_j)$.

**Data assumptions.** Addressing comments by reviewers QFiw and Yn8u, we clarified that exchangeability of tokens within each sequence is a sufficient condition for the coercivity in Lemma 3.4 (with a simple proof), but not a necessary one. Coercivity ensures that the learning problem is well-posed and may hold even when the data are not exchangeable. Moreover, the lower bound does not require exchangeability. Hence, our convergence results can be extended to general data distributions that satisfy the coercivity condition, regardless of exchangeability.


**Expanding the related works.** Following suggestions from reviewers xCfy and Yn8u, we expanded the related-work section to better connect our results with prior work on nonparametric and semiparametric works for neural networks (single and multi-index models, function composition and more), and added more references on non-softmax based attention models.

---

### Meta-Review · Area_Chair_QHae · 2026-01-04

**Summary:**

The authors studied a toy model of self attention in a low rank regression problem context, and provided minimax rates of learning such a problem based on the Holder exponent of the function.

The paper received two very positive reviews and two fairly negative reviews. Broadly speaking, the reviewers praised the submission for its technical details, with a fairly complete understanding of the regression problem with matching upper and lower bounds, and its non-triviality compared to the known single index problems studied previously.

On the other hand, the reviewers had two main criticisms:
 - The precise notion of dimension-free seems unsurprising or not novel
 - The problem being studied is too far away from practical self-attention

I will personally also add a third concern: I find the claim of "dimension-free" fairly misleading. If I were to only read the title and the abstract, I would not have expected a condition of the type $ M > (rd)^{2\beta+1}$. In fact, I would consider this result **not dimension-free**, as clearly that increasing dimension $d$ would also increase the number of samples $M$ required (once we are past a certain regime). As the authors mentioned, a more clear explanation of the error terms is split between

$$ M^{ \beta / ( 2\beta+1 ) } + \frac{rd}{M} $$

which if the authors complete a lower bound for the second term, the result can still be considered matching minimax rates. This is definitely my most significant concern for this submission.

However, I don't think the results are lacking novelty, or the problem being studied is too far from self-attention. I understand that two significant approximations are being used, but I can explain why neither are in my opinion problematic:
1. Avoiding the normalization term in the softmax. This is not problematic because it's not a key component of attention. There have been many existing work replacing the softmax with other non-linearities and have found essentially no difference in performance. The interpretation of the output summing up to 1 is simply not necessary in the same way that MLPs do not need sigmoid/tanh activations for their "interpretation" (we cannot really interpret these normalized outputs anyways).
2. Removing the value matrix. I also don't think this is problematic, because the value matrix behaves much closer to an MLP weight matrix, where as the attention component is multiplied on the token index instead of the width index. There have also been showing removing the value weight matrix does not hinder performance either.

If anything, the strongest assumption which no one criticized so far is the low rank part of the attention matrix. Although as far as a toy problem is concerned, in the spirit of multi-index models, this is considered an acceptable approach in attempt to separate complexity classes of the learning problem.

To summarize my concerns, I believe the claim of "dimension-free" is problematic, but otherwise the results are still very good. Especially in consideration of the recent events at ICLR with re-assigning area chairs, I believe this submission should still be accepted, **under the condition** that authors make **significant changes** to their title, abstract, and main claims of the results being dimension-free. For example, I would suggest an appropriate title could simply remove dimension-free and write "Minimax Rates for Learning Pairwise Interactions in Attention-Style Models". In the abstract and main claims, one could discuss a "dimension-free" term in the minimax rate instead of claiming the entire result to be dimension-free.

I hope the authors can understand my concern on why this result can appear misleading, and agree that the result are novel contributions even without the claim on being dimension-free.

**Reviewer Concerns:**

Dimension-free rate (xCfy, Yn8U, QFiw): clarified but not fully addressed.
On modelling attention (o395, xCfy, Yn8U): partially addressed, but in my opinion not a serious concern.
Lack of intuition (Yn8U and xCfy): mostly addressed.
Novelty and motivation (QFiw): partially addressed, other reviewers disagree.

**Reviewer Scores:**

xCfy: 8
Yn8U: 10
o395: 4 (potential raise to 5)
QFiw: 2 (maybe raise to 3, unlikely)

---

### Decision · Program_Chairs · 2026-01-26

Accept (Poster)